# Raman-guided subcellular pharmaco-metabolomics for metastatic melanoma cells

Jiajun Du[1,6], Yapeng Su [1,2,6], Chenxi Qian [1], Dan Yuan[2], Kun Miao [1], Dongkwan Lee[1], Alphonsus H. C. Ng[2], Reto S. Wijker [3], Antoni Ribas [4], Raphael D. Levine[5], James R. Heath [2✉] & Lu Wei [1✉]

Non-invasively probing metabolites within single live cells is highly desired but challenging. Here we utilize Raman spectro-microscopy for spatial mapping of metabolites within single cells, with the specific goal of identifying druggable metabolic susceptibilities from a series of patient-derived melanoma cell lines. Each cell line represents a different characteristic level of cancer cell de-differentiation. First, with Raman spectroscopy, followed by stimulated Raman scattering (SRS) microscopy and transcriptomics analysis, we identify the fatty acid synthesis pathway as a druggable susceptibility for differentiated melanocytic cells. We then utilize hyperspectral-SRS imaging of intracellular lipid droplets to identify a previously unknown susceptibility of lipid mono-unsaturation within de-differentiated mesenchymal cells with innate resistance to BRAF inhibition. Drugging this target leads to cellular apoptosis accompanied by the formation of phase-separated intracellular membrane domains. The integration of subcellular Raman spectro-microscopy with lipidomics and transcriptomics suggests possible lipid regulatory mechanisms underlying this pharmacological treatment. Our method should provide a general approach in spatially-resolved single cell metabolomics studies.

[1] Division of Chemistry and Chemical Engineering, California Institute of Technology, Pasadena, CA, USA. [2] Institute for Systems Biology, Seattle, WA, USA. [3] Division of Geological and Planetary Sciences, California Institute of Technology, Pasadena, CA, USA. [4] Department of Medicine, University of California Los Angeles, Los Angeles, CA, USA. [5] Department of Chemistry and Biochemistry, University of California Los Angeles, Los Angeles, CA, USA. [6] These authors contributed equally: Jiajun Du, Yapeng Su. ✉email: jheath@systemsbiology.org; lwei@caltech.edu

ingle-cell omics methods have revolutionized biology by resolving the heterogeneity that underlies population averages[1–5]. One envisioned application is that of pharmaco-omics (e.g., pharmacogenomics), in which the genetic or functional composition of diseased tissues is harnessed to guide the deployment of custom therapeutic strategies for individual patient[6,7]. Single-cell metabolomics has lagged behind other omics methods for the lack of proper toolsets for non-perturbative and targeted (analyte-specific) detection, but it has the potential to offer deep insights via shining light on the metabolic reprogramming that accompanies many disease states[8,9]. Mass spectrometry metabolomics has recently advanced to the level where analyte labeling techniques can permit multiplex analysis from single cells[10,11], but it is intrinsically sample-destructive, so prohibits live-cell analysis. The fluorescence-based methods offer high sensitivity[12], but with poor multiplexing, and fluorophore labels can hinder metabolite processing[8].

As a non-invasive optical tool, Raman spectroscopy probes the vibrational motions of chemical bonds, which allows detection of endogenous metabolites in a label-free manner. Multiple types of cellular metabolites have been identified by Raman fingerprinting, including nucleic acids, amino acids, lipids, glucose, neurotransmitters and etc[13–15]. In addition to spectroscopy, Raman microscopy further generates subcellular chemical maps by targeting predetermined vibrational peaks. In particular, the recent emergence of stimulated Raman scattering (SRS) microscopy, utilizing stimulated emission quantum amplification, provides imaging quality comparable to fluorescence microscopy with resolution of ~450 nm and speed up to video-rate in live cells and tissues[16,17]. By sweeping the laser across a designated wavelength range, hyperspectral-SRS (hSRS) rapidly produces Raman spectra of up to $600 \, \text{cm}^{-1}$ at subcellular locations[18–21]. Going beyond label-free analysis, Raman spectro-microscopy provides targeted detection and imaging of specific metabolites by recent strategies of stable-isotope labeling[22,23].

In this work, we explore Raman spectro-microscopy for subcellular pharmaco-metabolomics. We adopt a series of BRAF-mutant patient-derived melanoma cell lines as a model system. Metastatic melanoma is the most-deadly form of skin cancers, for which 66% of them harbor mutations in the BRAF kinase[24]. We utilize Raman spectro-microscopy to characterize this series of related but distinct BRAF-mutant melanoma cancer cell phenotypes, each corresponding to a different level of cancer cell differentiation, from melanocytic (differentiated) to mesenchymal (de-differentiated)[25–27]. The associated biology of these and similar melanoma models has been deeply investigated, which informs our study here[27–31]. The sensitivity of these cell lines to various targeted inhibitors and immunotherapies associates with de-differentiation status[27,28,32]. Differentiated phenotypes exhibit higher sensitivity toward BRAF inhibitors, while the de-differentiated phenotypes exhibit an innate resistance[27,33,34]. We hence mine the resulting spectroscopic information to identify phenotype specific, druggable metabolic susceptibilities.

We first establish a transcriptional relationship between cellular de-differentiation and metabolic reprogramming. We then integrate single-cell Raman data with transcriptomics analysis to establish that Raman-extracted trends in cellular chemical composition correlate with corresponding trends in gene expression. We identify and validate two druggable metabolic susceptibilities. One is specific to the differentiated melanoma cell lines studied, and is consistent with trends in gene expression. The second susceptibility is specific to the de-differentiated cell lines, and is uniquely extracted from the Raman analysis of subcellular lipid droplets (LDs). It is not detected through either bulk transcriptional analysis or bulk metabolomics, but can be validated by lipidomics. Raman analysis of single cells is thus shown as a potent pharmaco-metabolomics tool.

## Results

**Metabolic features are shown in transcriptome and Raman analysis.** Tsoi et al. recently published a pharmaco-genomic analysis of 53 patient-derived BRAF-mutant melanoma cell lines[27]. Notably, they demonstrated that the expression profiles of these cell lines faithfully reflected what was seen in the corresponding patient tumors. Further, they adopted unsupervised clustering of those profiles and classified the cell lines into four groups based upon de-differentiation status: melanocytic (differentiated), transitory, neural-crest-like, and mesenchymal (de-differentiated). We first selected a subset of 30 of these cell lines for analysis, on the basis that they did not also contain RAS mutations. Similar to reported[27], the whole transcriptomic data of these 30 cell lines, when visualized within a two-dimensional space (see "Methods"), yielded a clear separation into four distinct phenotypes, separated by level of de-differentiation (Fig. 1a, top panel). The nature of cancer cell de-differentiation means that energetic requirements, cellular morphology, etc., are all altered, suggesting that cellular differentiation is also accompanied by metabolic reprogramming[35]. We tested this hypothesis by similarly analyzing the same 30 melanoma cell lines, but including in that analysis only ~1600 genes associated with metabolic processes. In fact, this calculation yielded an almost identical clustering (Fig. 1a, bottom panel). Just like the well-reported phenotypic markers[28,36], metabolic genes also showed a clear phenotype-dependent expression trend, with associated functions that span different metabolic processes (The representative (top 4 ranked) metabolic genes are shown in the bottom of Fig. 1b, the complete heatmap and list of the top ranked metabolic genes are shown in Supplementary Fig. 1 and Supplementary Table 1). This implies that metabolic susceptibilities that exist within these cell lines may well vary with cellular de-differentiation, similar to what is known for inhibitors that target oncogenic signaling[37].

We selected five representative patient-derived cell lines based upon the single criteria that they collectively spanned the range of de-differentiation status (indicated at the top of Fig. 1b with information listed in Supplementary Table 2), from M381 (undifferentiated) to M262 (differentiated). We acquired spontaneous Raman spectra at the single-cell level from all five cell lines (Supplementary Fig. 2a) over the molecular fingerprinting spectral range of $700–3100 \, \text{cm}^{-1}$ (Fig. 1c). These spectral shapes are largely similar across four phenotypes. To extract differences between these spectra, we first utilized unsupervised surprisal analysis (SA) for dimension reduction[38]. SA is similar to principal component analysis (PCA) in that it is an orthogonal transformation of the data, with the dominant eigenvectors (also called constraints) capturing most of the variance observed in the Raman spectra of different cell lines. While SA has been successfully applied to analyzing gene expression datasets[26,39], an early application was for the analysis of molecular spectra[40]. We first confirmed that the constraints and their respective weights obtained from SA could recapitulate the fine Raman spectral features (Fig. 1d and Supplementary Fig. 2b). We then generated a heatmap of the top 5 constraints, labeled in ascending order as $\lambda_0$–$\lambda_4$, with each cell line represented by ten individual cells (Fig. 1e). The largest constraint, $\lambda_0$, captures universally shared spectral features and is expected to be invariant across cell lines. This shared spectrum, with peak assignments, is provided in Supplementary Fig. 3. The second largest constraint, $\lambda_1$ (Fig. 1e) captures the greatest variance from spectra to spectra and exhibits an average score that obviously changes with cellular de-differentiation (Fig. 1f). The remaining $\lambda_2$–$\lambda_4$ are lower in

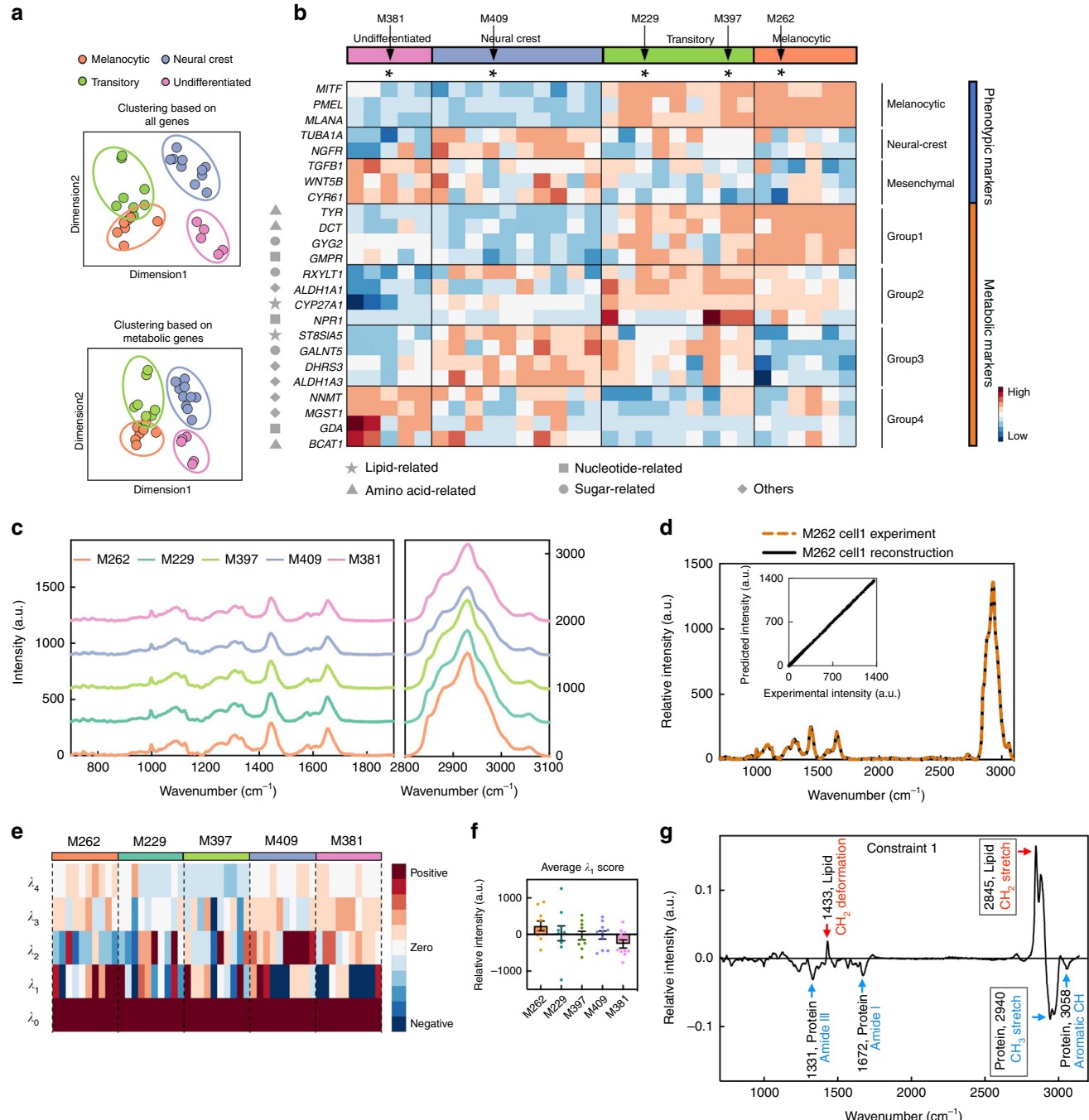

**Fig. 1 Transcriptomics and spontaneous Raman spectra analysis of metastatic melanoma cell lines. a** Dimensional reduction of bulk transcriptomics data of 30 melanoma cell lines yields a clear separation of four different melanoma phenotypes, based on either the expression of all genes (top panel) or ~1600 metabolic genes (bottom panel). **b** A heatmap of gene expression levels for representative genes involved in defining the cellular and metabolic phenotypes shown in **a**. The black-font row labels are well-reported phenotypic marker genes for defining different subtypes. The gray-font row labels are top 4 ranked metabolic markers within each phenotype representing different processes, as identified by matching the symbol with the key at the bottom of the heatmap. The color-coded bars at the top of the heat map indicate the different cellular phenotypes for each cell line, while the arrows point to the five representative cell lines selected for Raman analysis. **c** Spontaneous Raman spectra of five selected cell lines (averaged over 50 spectra from 10 cells per cell line examined over three independent experiments). Each spectrum is offset apart in $y$-axis with no changes of absolute intensities. **d** A representative Raman spectrum of M262 cells reconstructed by summing the constraints $\lambda_0$–$\lambda_4$ identified using surprisal analysis (SA). The inset plot shows the high correlation between the reconstructed and the measured spectrum. **e** Heatmap for scores of the top five constraints ($\lambda_0$–$\lambda_4$) calculated by SA of the Raman spectra across the five cell lines (10 cells from each cell line). Each column represents SA scores across $\lambda_0$–$\lambda_4$ from an individual cell. Each row represents the score of a given constraint across multiple single cells. **f** The average score of constraint 1 ($\lambda_1$) of 10 cells across all 5 cell lines. Data shown as mean ± SEM. **g** The spectrum of $\lambda_1$, with Raman peak assignments. The most negative feature is from $CH_3$ vibration at 2940 cm$^{-1}$ arising mainly from proteins (blue, boxed). The most positive feature is a $CH_2$ vibration at 2845 cm$^{-1}$ mainly from lipids (red, boxed). Source data are provided as a Source data file.

amplitude and less revealing in spectral features (Fig. 1e and Supplementary Fig. 3). The spectral distribution of $\lambda_1$ exhibits positive contributions from $CH_2$ vibrational stretches (2845 cm$^{-1}$, largely arising from lipids), and negative contributions from $CH_3$ stretches (2940 cm$^{-1}$, mostly from proteins) (Fig. 1g and Supplementary Fig. 4). The $\lambda_1$ score declines from M262 to M381. This indicates that the lipid/protein ($CH_2/CH_3$) ratio decreases with the progression of de-differentiation in these melanoma cell lines (Fig. 1g). We note here that the relative high variance in the $\lambda_1$ originates from the intracellular heterogeneity from relatively low sampling in spontaneous Raman acquisition (Supplementary Fig. 2a). This issue is largely bypassed in SRS imaging, as shown below, with much higher resolution and sampling.

**Differentiated cells are susceptible to fatty acid synthesis.** After mining the metabolic-associated spectral features from the wide fingerprint region, we next turned to live, single-cell imaging investigations to capture intracellular heterogeneity. We utilized SRS imaging (Supplementary Fig. 5a), with microsecond-level pixel dwell time, subcellular resolution and linear-concentration dependence for straightforward metabolic quantifications[16,17], to interrogate how the overall trend shown in Fig. 1f is reflected at the whole cell level. We targeted the lipid peak at 2845 cm$^{-1}$ (attributed to $CH_2$ vibrations[16,17], Fig. 2a, top) and the protein peak at 2940 cm$^{-1}$ (from $CH_3$ vibrations[41], Fig. 2a, middle). The generated $CH_2/CH_3$ ratiometric images (Fig. 2a, bottom) indeed nicely resolved a decreasing trend from melanocytic M262 cells toward mesenchymal M381 cells, implying that the more differentiated cells are relatively richer in lipids. SRS images on fixed cells yielded similar conclusions (Supplementary Fig. 5b). After quantifying the averaged $CH_2/CH_3$ intensity ratios (Fig. 2b and Supplementary Fig. 5c), we then asked whether this trend extracted from Raman imaging could be correlated to transcriptomics data. Strongly correlating or anti-correlating gene expression patterns are shown in the heatmap of Fig. 2c. In particular, several genes associated with lipid processing are identified with strong positive correlations, including fatty acid synthase (FASN), 3-hydroxyacyl-CoA dehydrogenase, and Malonyl CoA-acyl carrier protein transacylase, mitochondrial. In fact, the gene ontology (GO) fatty acid synthetic processes exhibits a strong linear correlation with the $CH_2/CH_3$ Raman ratios (Fig. 2d, top, $r = 0.93$, $p = 0.02$). Also notable are genes (Fig. 2c) and biological processes that exhibit a negative-correlation with $CH_2/CH_3$, such those associated with the cell migration pathway (Fig. 2d, bottom, $r = -0.91$, $p = 0.03$). The high migratory nature is a known feature in mesenchymal phenotypes[42]. Similar relationships from features strongly related to melanocytic (Supplementary Fig. 6a, top) or mesenchymal cell types (Supplementary Fig. 6a, bottom) were also resolved. These data demonstrate that single-cell Raman imaging yields information consistent with transcriptional profiling.

Elevated FASN expression (Supplementary Fig. 6b) in the differentiated cell lines implies increased de novo fatty-acid synthesis. We first sought to further explore this biology through targeted SRS imaging. Elevated glucose catabolism is a characteristic of many cancers, and produces an excess of the glycolytic end-product, pyruvate, some of which can be converted to acetyl-CoA and then further converted, through an FASN mediated pathway, to fatty acids[43,44] (Fig. 2e). The relative importance of de novo fatty-acid synthesis in the various cell lines can be inferred by tracking the conversion of glucose into fatty acids (Fig. 2e). Thus, we incubated the cells in media by replacing regular glucose with deuterated glucose (d$_7$-glucose) for 3 days before SRS imaging (Fig. 2f). The rationale is that an active de

novo fatty-acid synthetic pathway will convert some of this d$_7$-glucose into deuterated lipids, which exhibit a unique lipid associated C-D spectral signature around 2150 cm$^{-1}$, effectively yielding a live-cell assay of FASN activity[45]. SRS images of the five cell lines, collected at 2150 cm$^{-1}$, are provided in Fig. 2f. The measured cytoplasmic Raman spectrum (Supplementary Fig. 6c) matches what is expected from deuterated lipids[45]. The subsequent quantification of average C-D signals across multiples image sets (Fig. 2g) implies that de novo fatty acid synthesis is most activated in the differentiated cell lines M262, M229, and M397 and remains relatively low in de-differentiated M409 and M381.

Elevated FASN activities in the more differentiated melanoma cell lines suggest that the FASN pathway may constitute a metabolic susceptibility in just those phenotypes. In fact, interruption of this pathway has been previously studied for cancer drug development[46]. We tested this hypothesis by treating the cells with FASN inhibitors, 10 μM cerulenin[46] or 0.2 μM TVB-3166[47], for 3 days. As hypothesized, the three most differentiated phenotypes exhibited the highest sensitivity to cerulenin and TVB-3166 while the two most undifferentiated cell lines are barely affected by such drug treatments (Fig. 2h and Supplementary Fig. 6d). These data demonstrate that single-cell Raman spectro-microscopy, integrated with transcriptional profiling, can uncover phenotype-specific druggable susceptibilities in cancer cells.

**Mesenchymal M381 accumulates selected lipids in lipid droplets.** The above results indicate that metabolic susceptibilities within BRAF mutant melanoma cell lines can be strongly dependent upon de-differentiation phenotype. A second relevant example is that of mesenchymal-specific GPX4-inhibitor-induced ferroptosis identified using pharmacogenomics by Tsoi et al.[27]. That susceptibility is related to lipid peroxidation. Finding new druggable targets for the highly-invasive (Supplementary Fig. 7a) and BRAFi innate-resistant phenotype (Supplementary Table 2) might facilitate the development of clinically relevant inhibitors. We thus hypothesized that a deep interrogation of the lipid biochemistries in these cell lines might reveal additional druggable susceptibilities that distinguish the mesenchymal phenotypes. To this end, we studied the role of lipid storage in LDs. LDs are sub-micrometer-size lipid reservoir organelles[48,49] that are comprised of a highly dynamic mixture of neutral lipids (i.e., triacylglycerides (TAG) and cholesteryl esters (CE)). They are increasingly recognized for their central roles in modulating the transport and oxidation of lipids through interaction with other organelles[49,50].

We used hSRS microscopy to analyze the composition of these sub-cellular LDs at a spatial resolution of ~450 nm. Such live-cell compatible and non-perturbative subcellular quantification by hSRS is beyond what mass spectrometry and fluorescence analysis could offer. The unique spherical morphologies of LDs are readily imaged by SRS. Since they are lipid-rich, they exhibit large $CH_2$ Raman scattering signals near 2845 cm$^{-1}$ (Fig. 3a). We generated Raman spectra on LDs from each of the 5 cell lines, by acquiring SRS images across the C-H vibrational region from 2800 to 3050 cm$^{-1}$ with high spectral resolution of 8 cm$^{-1}$ (Supplementary Movie 1 and Fig. 3b). To extract the phenotype-dependent variations from these spectra, we again employed surprisal analysis (SA), which resolved a universal constraint $\lambda_0$, and just a single additional constant $\lambda_1$. As before, we confirmed that summing these two dominant constraints could recapitulate the measured hSRS spectra of LDs (Supplementary Fig. 7b). We then generated a heatmap of the weights of $\lambda_0$ and $\lambda_1$ for individual LDs, grouped by their associated cell lines (Fig. 3c). Again, $\lambda_0$ is

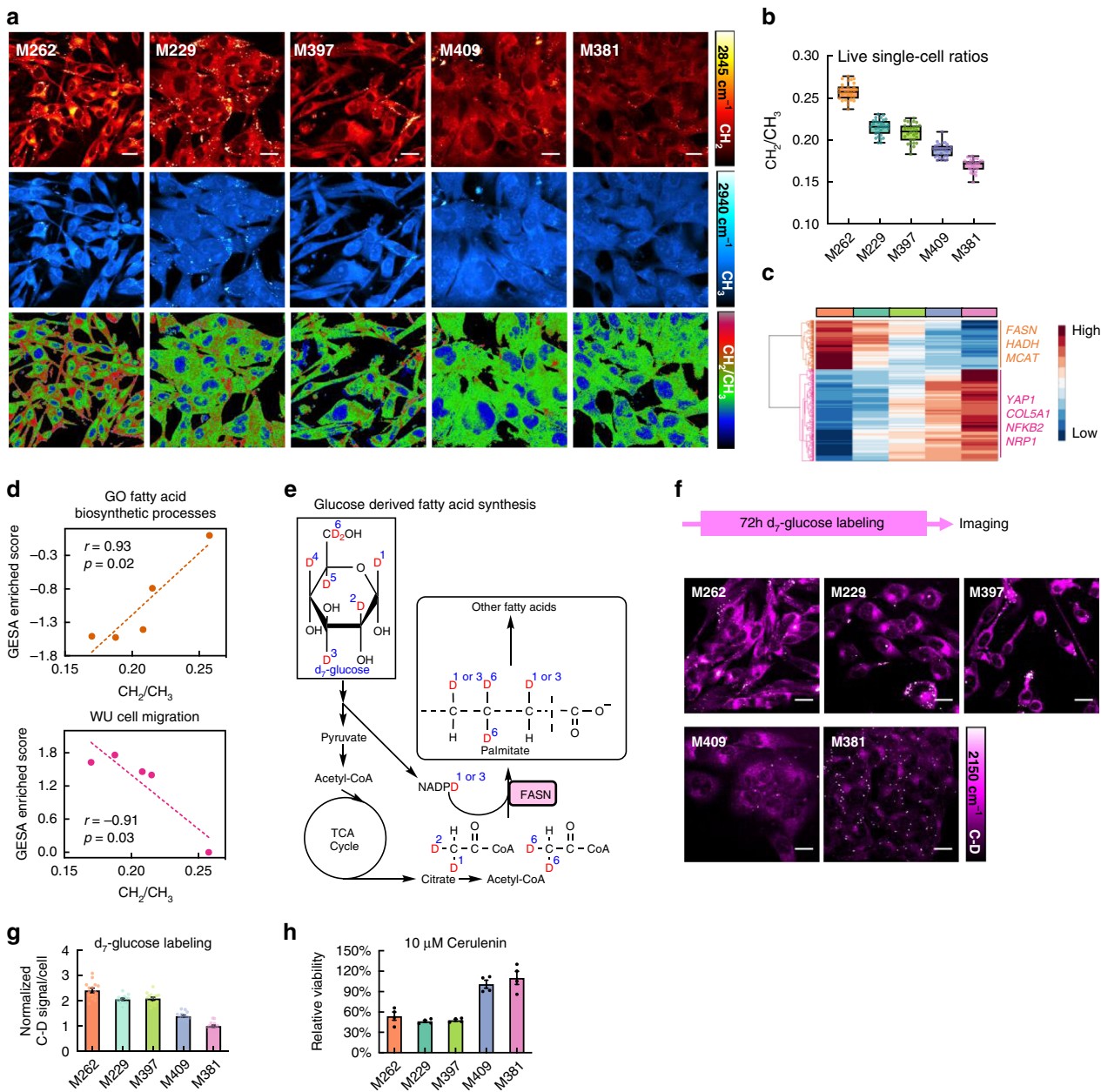

**Fig. 2 Live-cell SRS imaging and transcriptomics analysis reveal a differentiated-cell-specific susceptibility. a** Representative live-cell SRS images targeted on the $CH_2$ (top, 2845 cm$^{-1}$) and $CH_3$ (middle, 2940 cm$^{-1}$) channels and the corresponding $CH_2$ to $CH_3$ ratio (bottom, $CH_2/CH_3$) images. **b** Average live single-cell $CH_2/CH_3$ values from the SRS ratio images for each cell line ($n = 30$ cells per cell line examined over five independent experiments). Data are plotted as boxplots: center line indicates median; box limits indicate upper and lower quartiles; whiskers indicate minimum and maximum. **c** Heatmap of genes with strong correlations or anticorrelations to the $CH_2/CH_3$ trends shown in **b**. Representative genes involved in fatty acid metabolism (orange, positive correlation) and mesenchymal signature (purple, negative correlation) are indicated. **d** Two representative top biological functional processes from Gene Set Enrichment Analysis (GSEA) with GSEA scores that exhibit positive (top panel) or negative (bottom panel) correlations with the phenotype-dependent $CH_2/CH_3$ trends across different cell lines. **e** Illustration of the pathway for deuterium transfer from deuterated glucose ($d_7$-glucose) to de novo synthesized fatty acids through the major lipid biosynthetic pathways. **f** SRS imaging at the C-D channel (2150 cm$^{-1}$) for newly synthesized fatty acids in all 5 selected cell lines cultured with $d_7$-glucose medium for 3 days. Labeling and imaging scheme shown on top. **g** Single-cell quantification of relative C-D signals in $d_7$-glucose labeled cells ($n = 15$ cells examined over three independent experiments, the C-D signals of M381 cells are normalized to one). **h** Relative viability of melanoma cells after treatment of FASN inhibitor cerulenin (10 μM, 3 days, $n = 4$ independent experiments). Scale bars, 20 μm. Data shown as mean ± SEM. Source data are provided as a Source data file.

constant across cell lines (Fig. 3c and Supplementary Fig. 8a) while $\lambda_1$ exhibits a uniquely high positive amplitude for the mesenchymal M381 cell line (Fig. 3c, d). Based on Raman spectra from reference pure lipid species (Supplementary Fig. 8b), we annotated the spectral distribution of $\lambda_1$. The 3022 cm$^{-1}$ peak is assigned to the C–H stretch where the carbon is associated with a C = C

double bond (i.e., =C–H). This spectral feature arises mostly from unsaturated lipids (UL)[51]. The broad band from 2957 to 2997 cm$^{-1}$ largely originates from the C–H vibrations on the sterol rings of cholesterol ester (CE) (Fig. 3e)[52,53]. This spectral composition of $\lambda_1$ suggests that LDs within M381 cells bear the highest level of UL and CE among the five cell lines. This is further verified by

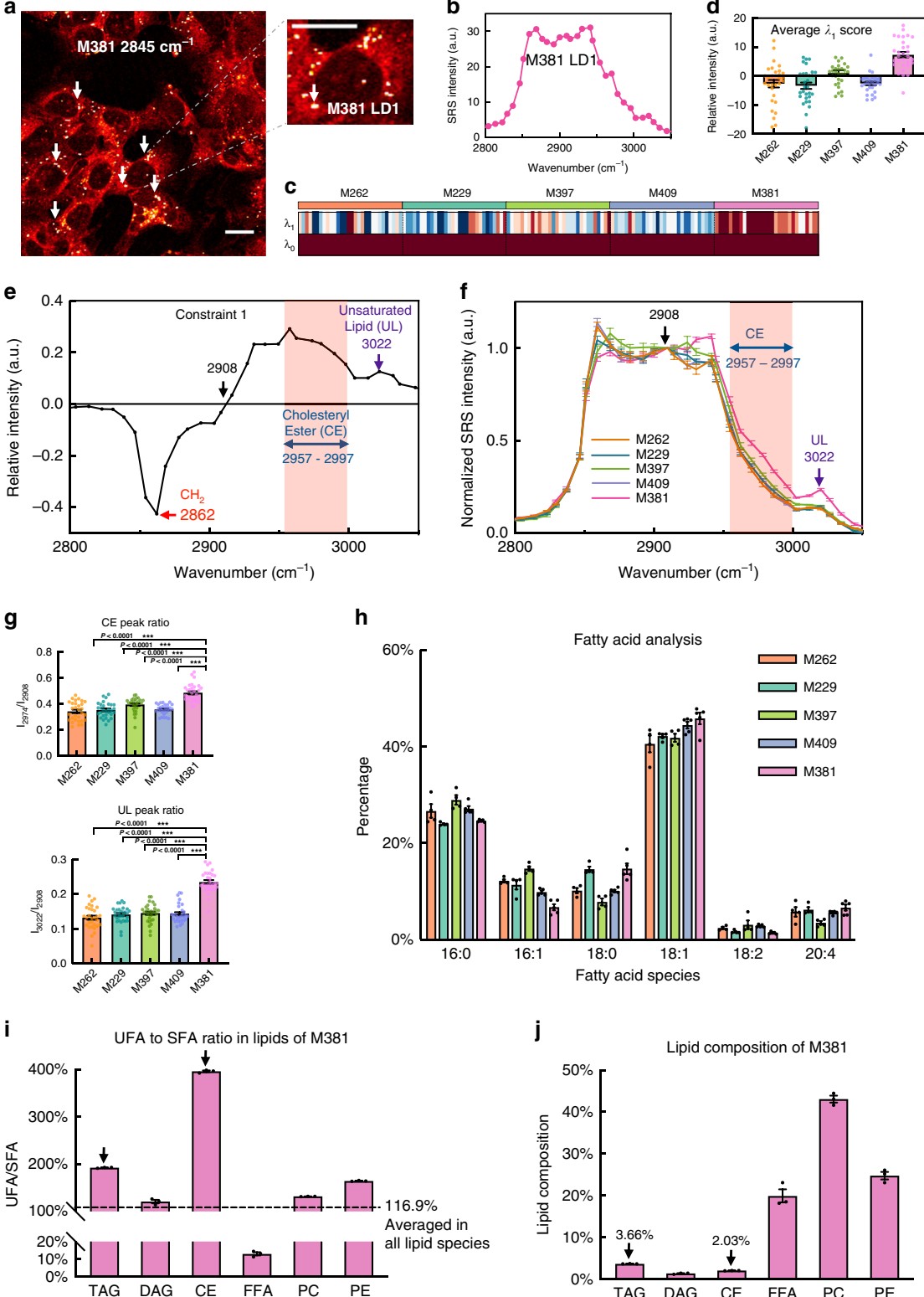

direct normalization of all hSRS spectra to 2908 cm$^{-1}$ (Fig. 3f, g), which is a zero point in $\lambda_1$ (Fig. 3e).

The observation that the intracellular LDs within the mesenchymal M381 cell line exhibit a relatively increased level of unsaturated lipids, relative to the other cell lines (Fig. 3g, bottom), suggested a novel lipid regulation process within that cell line. We first examined whether this trend of lipid unsaturation was reflected in bulk analysis. We performed gas

chromatography-mass spectrometry (GC-MS) based analysis of fatty acids from cell pellets (Fig. 3h). For presentation, we follow the common lipid notation of xx:yy, where xx represents the number of carbon atoms in the lipid chain, and yy refers to the number of double bonds (Fig. 3h). Although M381 cells show slightly enhanced level of 18:1 fatty acid (i.e., oleic acid) relative to the other cell lines, the heterogeneity of overall unsaturation across the cell lines is minor. Similarly, there is no clear trend for

**Fig. 3 Accumulation of unsaturated lipids and cholesteryl esters in lipid droplets (LDs) of de-differentiated M381 cells. a** A representative SRS image of M381 cells imaged in the $CH_2$ (2845 $cm^{-1}$) channel. LDs are indicated. A zoomed-in image at right highlights a single LD. **b** The hSRS spectrum of the zoomed-in LD in **a** at the C–H stretch region (2800–3050 $cm^{-1}$). **c** Heatmap for scores of the top two constraints ($\lambda_0$–$\lambda_1$) by surprisal analysis of hSRS spectra on LDs across five cell lines ($n = 30$ LDs per cell line examined over three independent experiments). Each column represents an individual LD and each row represents the constraint scores. **d** The average score of $\lambda_1$ across five cell lines ($n = 30$ LDs). **e** Raman peak assignments for constraint 1 ($\lambda_1$). The pink shadowed range from 2957 to 2997 $cm^{-1}$ is assigned to cholesteryl esters (CE), and the 3022 $cm^{-1}$ peak (violet arrow) is assigned to unsaturated lipids (=C–H, UL). **f** hSRS spectra (normalized at 2908 $cm^{-1}$, the zero point revealed in **e**) of LDs across each cell line ($n = 30$). **g** Quantification of relative CE (2974 $cm^{-1}$/2908 $cm^{-1}$, top panel) and UL (3022 $cm^{-1}$/2908 $cm^{-1}$, bottom panel) enrichment in LDs across cell lines from **f** ($n = 30$). **h** GC-MS measurement of fatty acids extracted from bulk melanoma cells. The percentages of 16:0, 16:1, 18:0, 18:1, 18:2, and 20:4 are normalized to all extracted fatty acids ($n = 4$ independent experiments for M262, M229, M397; $n = 5$ independent experiments for M409 and M381). **i** Average ratio of unsaturated fatty acids (UFA) to saturated fatty acids (SFA) in each lipid class from lipidomics of M381 cells ($n = 3$ independent experiments). **j** Percentage of major lipid classes from lipidomics of M381 cells ($n = 3$). ***$p < 0.001$ from two-tailed unpaired $t$-tests. Scale bars, 20 μm. Data shown as mean ± SEM. Lipidomics data are provided as Supplementary Data 1. Source data are provided as a Source data file.

the expression levels of key desaturases in M381 compared to other cell lines (Supplementary Fig. 8c). It is likely that the compositional variability of neutral lipids (i.e., TAG and CE) in LDs is averaged out by other more abundant lipid species in the bulk GC-MS analyses. Therefore, we performed liquid chromatography-mass spectrometry (LC-MS) based lipidomics profiling with preserved lipid structures. Indeed, bulk lipidomics data for M381 cells clearly shows that while droplet-enriched species of TAG and CE have the highest unsaturated fatty acid (UFA) composition among major lipid species (Fig. 3i), they only account for a small portion (in total <6%) of all major lipid species (Fig. 3j). Thus, M381 has elevated lipid unsaturation levels specifically within intracellular LDs.

**Desaturases are involved lipid-droplet unsaturation of M381.** We next sought to trace the source of the enhanced lipid-droplet unsaturation in M381 cells. Such an increase may arise from either cellular uptake or de novo synthesis. Further, the unsaturated lipid signal could originate from either mono-unsaturated fatty acids (MUFA) or poly-unsaturated (multiple double bonds) fatty acids (PUFA). First, to assess lipid uptake, we adopted a labeled SRS imaging approach by incubating M381 cells in medium containing deuterated MUFA ($d_{33}$-oleic acid) or saturated fatty acids (SFA) ($d_{31}$-palmitic acid), the two most widely used fatty acids for assaying uptake. We found that M381 cells have the lowest uptake of extracellular fatty acids across all cell lines (Supplementary Fig. 9), suggesting that de novo synthesized fatty acids may serve as major sources for M381. We next tested whether the MUFA or PUFA de novo synthesis pathway (Fig. 4a) contributes to the elevated lipid-droplet unsaturation. In mammalian cells, Δ9 desaturase (Stearoyl-CoA desaturase-1, SCD1) is the rate-limiting enzyme for MUFA generation, specifically for producing oleic acids (OA, 18:1) and palmitoleic acids (PO, 16:1) from stearic (ST, 18:0) and palmitic (PA, 16:0) acids (Fig. 4a). In addition, Δ6 and Δ5 desaturases contribute to generating functionally important PUFA, such as docosahexaenoic acid (DHA, 22:6) and arachidonic acid (AA, 20:4) by catalyzing the formation of additional double-bonds from essential fatty acids of linoleic acid (LA, 18:2) and alpha-linolenic acid (ALA, 18:3) (Fig. 4a). We adopted pharmacological approaches to probe these pathways. CAY10566 (CAY) and SC 26196 (SC) are Δ9 (SCD1) and Δ6 desaturase inhibitors, respectively (Fig. 4a)[51]. Upon treatment with varying doses of CAY or SC on M381 for 3 days, our hSRS spectra revealed decreasing levels of unsaturation within LDs (Fig. 4b, c, 3022 $cm^{-1}$), demonstrating the involvement of both MUFA and PUFA in LDs. This spectral response for decreased unsaturation upon drug treatment was also well-reflected in the heatmap of constraint scores by SA (Supplementary Fig. 10). In addition, the involvement of MUFA and PUFA in LDs of M381

was supported by lipidomics of TAG and CE (Supplementary Fig. 11), the main LD species.

**Inhibiting SCD1 but not Δ6 desaturase induces apoptosis in M381.** Although both CAY and SC reduce the lipid-droplet unsaturation levels, CAY inhibition of SCD1 for MUFA synthesis leads to a more significant loss of viability for M381 cells relative to the other four cell lines (Fig. 4d and Supplementary Fig. 12), while SC treatment to block the Δ6 desaturase for PUFA synthesis pathway barely affects the viability of M381 (Fig. 4e). It is worth noting that this specific susceptibility of SCD1 in M381 is not indicated by bulk gene expression of SCD1 (Supplementary Fig. 8c) or bulk fatty acid analysis (Fig. 3h). The inhibitory function of SCD1 is further confirmed by small hairpin RNA (shRNA) based gene silencing of SCD1 (Fig. 4f). This result illustrates that SCD1 inhibition could be a susceptibility of mesenchymal M381 cells and inspired us to develop a deeper understanding of SCD1 regulation in M381 cells. First, our bulk GC-MS analysis on major fatty acid species from cell pellets showed that SCD1 inhibitor mostly blocks the generation of the monounsaturated OA (18:1) from saturated ST (18:0) (Fig. 4g). This is consistent with the knowledge that OA (18:1) is the principle product of SCD1[54]. Second, a time-lapse apoptosis video assay demonstrated that CAY reduces the viability of M381 by inducing apoptosis (Fig. 4h). Surprisingly, both the time-lapse apoptosis (Fig. 4h) and the time-dependent viability assays (Fig. 4i) revealed that the M381 cells do not initiate apoptosis program until 1–2 days treatment with CAY. A similar lagging effect is also observed for the decrease in the hSRS spectral signature for unsaturation within LDs (Fig. 4j, k). Taken together, the GC-MS and the kinetics data imply that the susceptibility of CAY may originate from the gradual depletion of OA (18:1) and/ or the corresponding accumulation of ST (18:0).

**SCD1 inhibition induces phase-separated membrane structures.** Lipotoxicity from excessive SFA (e.g., PA, 16:0 and ST, 18:0) is a well-documented effect that impairs cellular functions by inducing endoplasmic reticulum (ER) stress[55–57], unfolded protein response (URP)[55–57] and the formation of ceramides[55] and reactive oxygen species[57]. Recently, it was found in live HeLa cells that supplying extra SFA into the culture medium could convert the intracellular membranes from the regular liquid-disordered phase into an ordered-solid phase[58]. This resulted in perturbed membrane functions and induced cell death. The conversion of a fluidic normal membrane (NM) into a rigid solid membrane (SM) can be characterized by detergent wash, in which the NM will be removed while the SM is not[58]. Since CAY treatment of M381 mostly reduces intracellular OA (18:1) while increasing ST (18:0) levels by blocking the ST-to-OA conversion (Fig. 4a, g), we

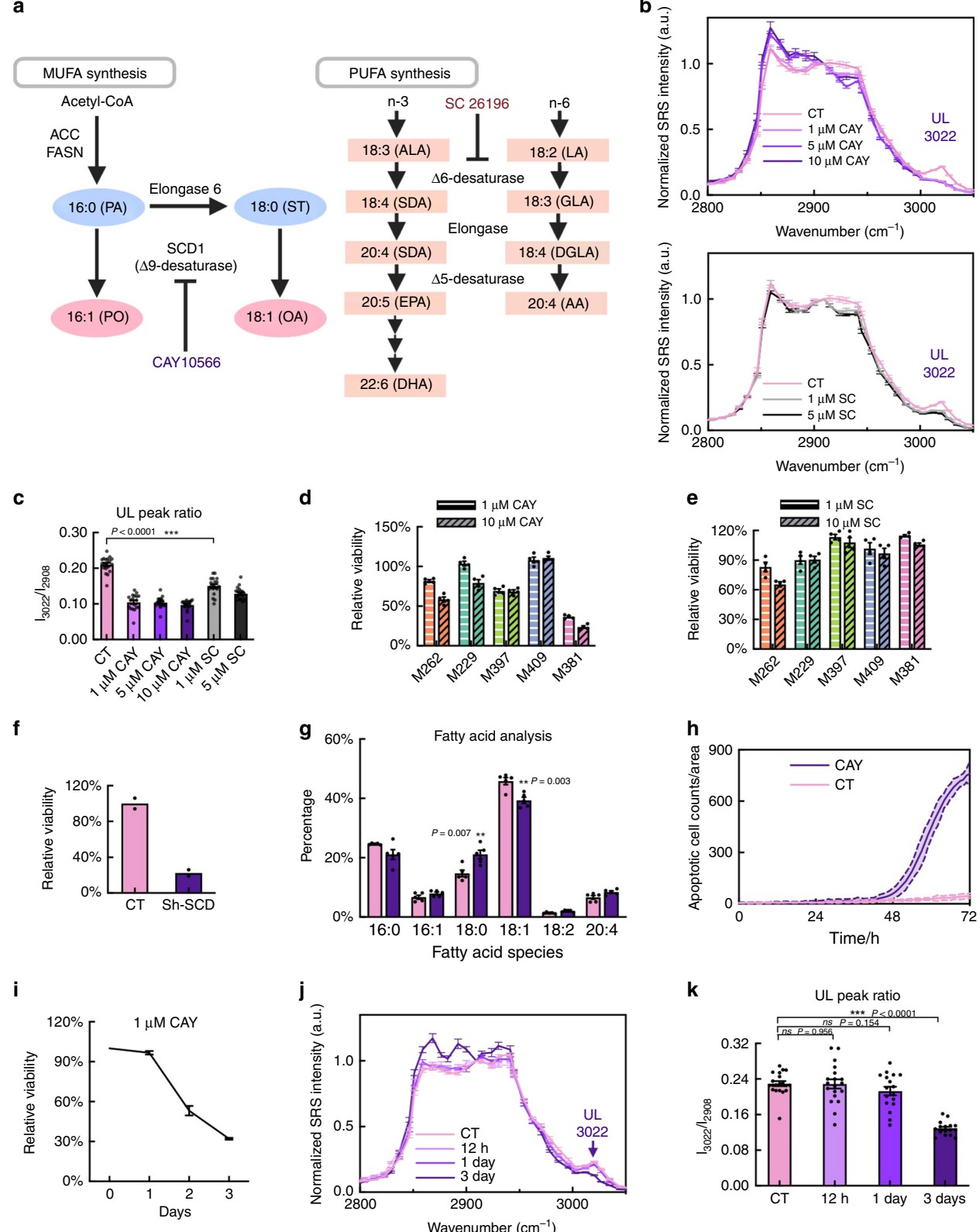

hypothesized that the resulting lipid imbalance would lead to phase-separated SM domains.

We used SRS imaging to characterize potential phase changes in M381 cell membrane after CAY treatment. Indeed, we observed that the membrane structures were clearly extracted by detergent wash in both control (CT) cells and cells treated with the Δ6 desaturase inhibitor SC (1 μM and 5 μM for 3 days) (Fig. 5a before

vs after wash), indicating that these cells contain NM. By comparison, for cells treated with CAY (1 μM and 5 μM for 3 days), the membrane structures were detergent-resistant, indicating the conversion of NM to SM structures (Fig. 5a, before vs after wash). We next characterized the composition change of SM in situ by comparing the hSRS spectra on selected intercellular regions of detergent-extractable NM, detergent-resistant SM

**Fig. 4 SCD1-dependent viability for the mesenchymal M381 cells. a** (Left) De novo synthesis pathway of monounsaturated fatty acids (MUFA) and (right) polyunsaturated fatty acids (PUFA) in mammalian cells. CAY10566 and SC 26196 are SCD1 ($\Delta$9-desaturase) and $\Delta$6-desaturase inhibitor, respectively. **b** Normalized (to 2908 cm$^{-1}$) hSRS spectra of LDs in M381 cells without (CT) and with treatment of (top) 1, 5 and 10 $\mu$M CAY ($n = 16, 18, 19, 19$ for CT, 1, 5 and 10 $\mu$M CAY, respectively), and (bottom) 1 $\mu$M and 5 $\mu$M SC ($n = 16, 19, 19$ for CT, 1 $\mu$M and 5 $\mu$M SC, respectively) for 3 days. **c** Quantification of unsaturated lipid (UL) by intensity ratios of 3022 cm$^{-1}$/2908 cm$^{-1}$ from **b**. Relative viability of all five cell lines after treatment of 1 $\mu$M and 10 $\mu$M CAY for 3 days ($n = 4$ independent experiments) (**d**) or 1 $\mu$M and 10 $\mu$M SC for 3 days ($n = 4$ independent experiments) (**e**). **f** Relative viability of M381 cells after shRNA knockdown of *SCD1* gene compared to scrambled control (CT) ($n = 2$ independent experiments). **g** GC-MS measurements of fatty acids extracted from bulk M381 cells with (CAY, purple) and without (CT, pink) treatment of 1 $\mu$M CAY for 3 days. The percentages of 16:0, 16:1, 18:0, 18:1, 18:2, and 20:4 fatty acids are normalized to total extracted fatty acids ($n = 5$ independent experiments). **h** Time-lapse apoptotic cell counts of M381 cells with (purple, CAY) and without (pink, CT) treatment of 1 $\mu$M CAY ($n = 3$ independent experiments, data shown as mean ± error with 95% CI). **i** Time-dependent relative viability of M381 cells after treatment of 1 $\mu$M CAY for 0, 1, 2, and 3 days ($n = 4$ independent experiments). **j** Normalized (to 2908 cm$^{-1}$) hSRS spectra of LDs in M381 cells without (CT) and with 1 $\mu$M CAY treatment for 12 h, 1 day and 3 days ($n = 20, 19, 17, 14$ for CT, 12 h, 1 day and 3 days, respectively). **k** Quantification of UL from intensity ratios of 3022 cm$^{-1}$/2908 cm$^{-1}$ in **j**. **\*\***$p < 0.01$, **\*\*\***$p < 0.001$, ns: not significant ($p > 0.05$) from two-tailed unpaired *t*-tests. Data shown as mean ± SEM. Source data are provided as a Source data file.

domains, and LDs (Fig. 5b, normalized to 2908 cm$^{-1}$ as previously indicated). First, comparing the spectra of NM from before and after wash, the greatly reduced peak at 2845 cm$^{-1}$ confirmed the effective extraction of most lipid contents by detergent wash (Fig. 5b, blue solid line vs blue dashed line). The maintained intensity at 2940 cm$^{-1}$ after wash suggests that the NM is also enriched with proteins. In contrast, both SM and LD exhibit a largely maintained SRS spectral shape following detergent treatment (Fig. 5b, red, SM before vs after and green, LD before vs after). This indicates that both structures are resistant to detergent wash. Interestingly, the overall Raman spectral shape of SM is very distinct from that of NM (Fig. 5b, NM before vs SM before), but is similar to that from the LDs (Fig. 5b, SM before vs LD before). The similarity indicates that the SM is highly lipid-rich. The difference between SM and NM suggests that the formation of phase-separated SM domains causes an exclusion of membrane-residing proteins, consistent with previous models that proteins or peptides which are anchored in intracellular membranes by $\alpha$-helix clearly prefer the liquid phase and would be excluded by the solid phase for dimerization[56,59]. Thus, CAY inhibition on M381 cells indeed causes the formation of phase-separated intracellular solid-membrane structures that enrich lipids, but exclude proteins.

Since CAY inhibition of SCD1 affects the de novo fatty-acid synthesis pathway (Fig. 4a), the SMs should have a high accumulation of newly synthesized lipids. Having identified that de novo lipid synthesis in our melanoma cells traces back to glucose (Fig. 2e, f), we again supplied M381 cells with d$_7$-glucose, but this time together with CAY treatment for 3 days. We then used SRS imaging of C–D vibrations at 2150 cm$^{-1}$ to visualize lipids that are synthesized specifically during the treatment period. As expected, we detected a formation of SM structures that were retained after detergent wash from C–D SRS images, which show similar patterns to that in the C–H channel (Fig. 5c, before vs after wash). This observation confirms that the newly-synthesized saturated lipids contribute to the formation of SM upon SCD1 inhibition of CAY.

CAY treatment induces the formation of SM structures by blocking the cellular conversion of newly synthesized SFA to UFA. This imbalance of homeostasis between SFA and UFA may also be caused by supplying cells with extra amount of SFA in the medium, which could promote the formation of SM structures[58]. Indeed, we observed the appearance of solid-membrane patterns by treating M381 cells with 100 $\mu$M deuterated palmitic acid (PA, 16:0) or 50 $\mu$M deuterated steric acid (ST, 18:0) (Fig. 5d, before vs after). Interestingly, the viability assays with PA and ST treatment (Fig. 5e) exhibited a M-shaped trend across all 5 cell lines. This trend is similar to that with CAY treatment (Fig. 4d), suggesting a similar toxicity effect between CAY and SFA. As a control, incubating cells with extra UFA show negligible toxicity for all cell lines (Supplementary Fig. 13a, PO, 16:1 and OA, 18:1). In addition, in similar ways the invasiveness of M381 cells is impaired with either CAY, PA or ST treatments (Supplementary Fig. 13b). The loss of invasiveness is likely because the formation of SM structures leads to a loss of membrane fluidity, which is required for metastatic cancer cells to invade through the dense basement membrane[60]. We again validated the formation of intracellular solid-membrane structures and their associated cytotoxicity when cellular pool of SFA exceeds that for the homeostatic level in M381 cells.

**Lipidomics suggest a reservoir role of LDs.** To obtain a comprehensive picture of how SCD1 inhibition perturbs lipid homeostasis, we carried out bulk lipidomics analysis of M381 cells with and without CAY treatment (Fig. 6a). We presented heatmaps on six major intracellular lipid species of TAG, diacyglycerol (DAG), CE, free fatty acid (FFA), phosphatidylcholine (PC) and phosphatidylethanolamine (PE) in control, and CAY-treated cells (Fig. 6a). Heterogeneous remodeling for SFA and UFA is revealed across different lipid species (Fig. 6a, pink, SFA, saturated fatty acids; green, UFA, unsaturated fatty acids of different acyl chain length and double-bond number). For quantification, we first plot the ratios of total SFA to total UFA in each of six lipid species for control cells (CT, pink) and CAY-treated (1 $\mu$M, 3 days, purple) cells (Fig. 6b, SFA/UFA). Indeed, SFA/UFA ratios increase, although to a different extent, across all six lipid species in CAY-treated M381 samples. The increase is particularly obvious for TAG and CE (Fig. 6b), the main residents in LDs. This again explains why hSRS spectroscopy and imaging on LDs is so revealing. The difference between CT and CAY-treated cells becomes particularly obvious for the ST/OA (i.e., 18:0/18:1) ratios in each species (Fig. 6b). This is consistent with our previous GC-MS results (Fig. 4g) that the function of SCD1 is more strongly directed toward the generation of OA (18:1) from ST (18:0) relative to the generation of PO (16:1) from PA (16:0) (Supplementary Fig. 13c). Further, the total concentrations of membrane lipids, PC and PE, increase by 40.2% and 38.6% after CAY treatment (Supplementary Fig. 13d), which may explain the abnormally high lipid signals observed in SMs (Fig. 5a). These species-dependent lipidomics heatmaps and ratio analysis confirm the relative increase of saturation level across all different lipid species and identify the more dominant changes in both TAG and CE under CAY treatment, consistent with our SRS data.

Further quantification of the absolute concentrations of ST (18:0) and OA (18:1) from lipidomics (Fig. 6c) yields additional mechanistic insights into cellular behaviors under CAY inhibition. First, under CAY inhibition that blocks the conversion of ST

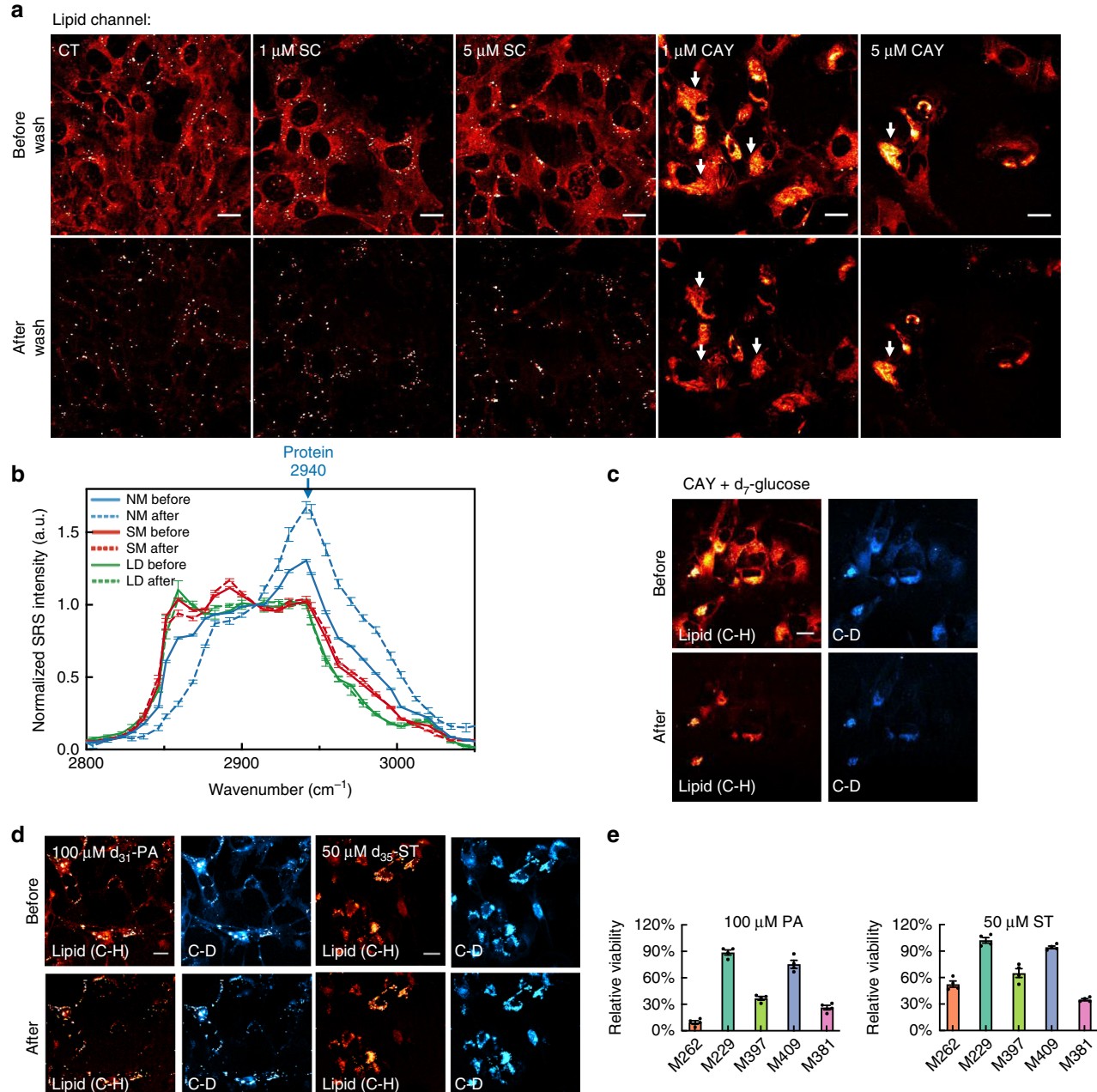

**Fig. 5 Formation of intracellular phase-separated solid membrane (SM) domains induced by mono-unsaturation inhibition. a** Representative lipid-channel SRS images from the same set of M381 cells before (top) and after (bottom) detergent wash in control (CT), with SC or CAY treatment. Detergent-resistant SM structures are arrow indicated. **b** Normalized (to 2908 cm$^{-1}$) hSRS spectra on the same NM (normal membrane, blue), SM (red), and LDs (green) structures in M381 cells from before (solid-lined) and after (dash-lined) detergent wash. ($n = 7$, 12, 6 for NM, SM, LD, respectively, blue arrow indicates the protein peak at 2940 cm$^{-1}$). **c** SRS images at the lipid (C-H) and the C-D channels on the same set of M381 cells growing in d$_7$-glucose medium with 1-day 5 μM CAY before (top) and after (bottom) detergent wash. **d** SRS images at the lipid (C-H) and the C-D channels on the same set of M381 cells with 3-day d$_{31}$-palmitic acid (d$_{31}$-PA) or d$_{35}$-stearic acid (d$_{35}$-ST) treatment before (top) and after (bottom) detergent wash. **e** Relative cellular viability with 3-day PA or ST treatment ($n = 4$ independent experiments). Scale bars, 20 μm. Data shown as mean ± SEM. Source data are provided as a Source data file.

(18:0) to OA (18:1), DAG, CE, FFA, PC, and PE all show statistically significant increases in the levels of ST (18:0). This is likely due to the continuous synthesis and accumulation of new ST. Interestingly, the increase of ST (18:0) in TAG is not statistically significant. On the other hand, TAG presents a large drop in OA (18:1) level upon CAY treatment, while the other five lipid species have an approximately unchanged OA (18:1) level. This suggests that OA (18:1) in TAG may be hydrolyzed and released under CAY treatment. Taken with our previous kinetic

SRS data that the unsaturation levels of LDs, which are mainly comprised of TAG and CE, only decrease after 1-day of CAY treatment (Fig. 4j, k), we suggest a possible reservoir role of TAG for UFA in the LDs of M381 cells. After SCD1 inhibition blocks the conversion of newly synthesized ST (18:0) to OA (18:1), the cytosolic saturation level increases. When the level of newly synthesized SFA in the cytosol reaches a threshold (in our case, after 1-day of CAY treatment), the TAG in the LDs starts to release UFA (e.g., OA) to restore the balance of cellular lipid

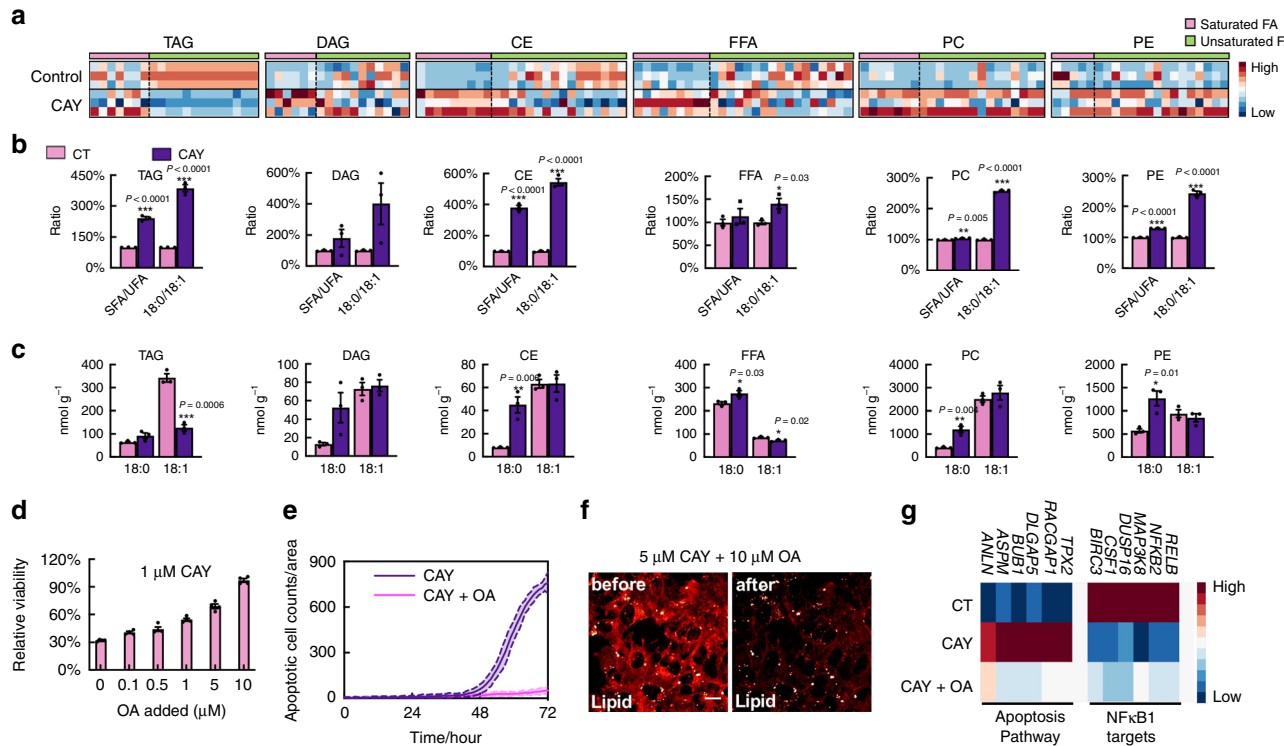

**Fig. 6 Lipid regulation upon mono-unsaturation inhibition and cellular rescue with oleic acid (OA) supplementation. a** Heatmap for the relative fatty acid abundances in designated classes of lipids from lipidomics of bulk M381 cells with (CAY) and without (CT) CAY (1 μM, 3 days) treatment. Each row represents a sample replicate and each column represents fatty acyl chains with increasing length and increasing double-bond numbers. SFA and UFA are categorized by pink and green, respectively. The abundance of FA of different chain length is normalized as a Z-score across all six samples within each column. TAG Triacylglycerol, DAG Diacylglycerol, CE Cholesteryl Ester, FFA Free Fatty Acids, PC Phosphatidylcholine, PE Phosphatidylethanolamine. **b** Overall concentration ratios between SFA to UFA (SFA/UFA), and between ST to OA (18:0/18:1) with CAY (purple) and without CAY (CT, pink, each value is shown as 100% reference) treatment from lipidomics of M381 cells in **a** ($n = 3$). **c** Concentration changes of 18:0 and 18:1 with CAY (purple) and without CAY (CT, pink) from lipidomics in **a** ($n = 3$ independent experiments). **d** Relative viability of M381 cells treated with 1 μM CAY with supplement of OA (18:1) at indicated concentration for 3 days ($n = 4$ independent experiments). **e**, Time-lapse apoptotic cell counts of M381 cells after treatment of 1 μM CAY with (pink) or without (purple) 10 μM OA ($n = 3$ independent experiments, data shown as mean ± error with 95% CI). **f** SRS imaging at the lipid channel before (left) and after (right) detergent wash on the same set of M381 cells treated with CAY and OA for 3 days. **g** Ranked pathways from analysis of gene expression trends on control (CT), CAY-treated (middle row, CAY) and CAY plus OA (CAY + OA) M381 cells. Scale bars, 20 μm. Data shown as mean ± SEM. *$p < 0.05$, **$p < 0.01$, ***$p < 0.001$ from two-tailed unpaired $t$-tests. Lipidomics data are provided as Supplementary Data 1. Source data are provided as a Source data file.

unsaturation. With the continuous depletion of UFA from TAG in LDs under prolonged CAY treatment, this storage is eventually depleted. The imbalance of intracellular SFA/UFA ratios then leads to the formation of toxic SM structures, as observed in Fig. 5a.

**Supplying UFA rescues SCD1-induced apoptosis in M381**. We reasoned that supplying CAY-treated cells with extra UFA, such as OA, may rescue the toxicity effect of the drug by restoring the balance between SFA and UFA. Indeed, both cell viability (Fig. 6d) and cell invasiveness (Supplementary Fig. 13e) of CAY-treated M381 cells were restored by adding OA in the medium together with CAY in a dose-dependent manner. Our time-lapse apoptotic assay (Fig. 6e) confirmed that high-dose (10 μM) of OA fully rescues M381 cells from apoptosis under CAY (1 μM) treatment. Further, with co-treatment of OA and CAY, the phase separated solid-membrane structures are absent, even at higher concentration (5 μM) CAY (Fig. 6f, before vs after). It is known that OA supplementation can reduce lipotoxicity by channeling extra cytosolic SFA into LDs[61]. We hence performed a pulse-chase experiment to explore the possible rescue effect (Supplementary Fig. 13f). We first pulse-treated M381 cells with 5 μM CAY for 60 h. Verified the formation of solid-membrane

structures in this condition (Supplementary Fig. 13f, lipid, C-H), we then chased (i.e., rescued) the cells with 20 μM of deuterated OA ($d_{33}$-OA) for another 10 h. We observed much less solid-membrane (Supplementary Fig. 13f, set 1 and 2, C-H) and a significantly increased number of LDs derived from deuterated OA (Supplementary Fig. 13f, set 1 and 2, boxed, C-D). We also queried whether other UFA could have a similar rescue effect. At a low dose (1 μM), OA is the most effective tested rescue agent (Supplementary Fig. 13g). This may be due to the preference of the OA substrate by the key enzymes, Diacylglycerol O-Acyltransferase 1 and Diacylglycerol O-Acyltransferase 2, involved in TAG formation[62,63]. At higher concentration (5 μM), other UFA (PO, 16:1; LA, 18:2; AA, 20:4) can reach similar rescue (Supplementary Fig. 13h), showing that the key is to restore the cellular balance between SFA and UFA.

To understand specific gene regulatory pathways involved in the saturated-lipid associated M381 susceptibility, we carried out RNA-seq transcriptomics analysis on CT cells, cells treated with 1 μM CAY (CAY), and cells co-treated of 1 μM CAY and 1 μM OA (CAY + OA). We ranked the gene sets that either exhibit increased or decreased expression levels under CAY treatment relative to CT, and then exhibit restoration under CAY + OA treatment. Two pathways stand out (Fig. 6g). First, the apoptosis

pathway (Fig. 6g, left and middle columns) is upregulated with CAY treatment and recovers with CAY + OA. This observation is consistent with our functional assays (Figs. 4h, 6e). Second, the NFκB1-targets pathway exhibits decreased expression with CAY and recovers with CAY + OA. The high NFκB transcriptional state in melanoma has been suggested to be BRAFi resistant, consistent with what is known for M381[25,27,64]. In addition, the NFκB pathway has been implicated in maintaining the stemness feature in ovarian cancer stem cells[51], so it might play a similar role here in maintaining the mesenchymal nature of M381. In previous reported lipotoxicity studies, perturbation of cellular lipid composition through the use of either relatively high concentrations (~0.5 mM) saturated lipid in the culture medium, or via SCD1 inhibitors, was shown lead to the activation of ER stress sensors and the UPR[55–57]. In this study, transcriptional signatures associated with neither ER nor UPR stresses were significantly elevated following 1 μM CAY treatment (Supplementary Fig. 13i). One possibility is that suppression of protein translocation into the SM structures may trigger proapoptotic signaling.

## Discussion

Single-cell metabolomics is challenging because there is neither a general amplification strategy, such as PCR, nor a general capture agent approach, such as antibodies, to facilitate the detection of specific metabolites with required sensitivity. Here, we demonstrated that Raman spectro-microscopy opens up the ability to spatially resolve and quantitatively analyze particular classes of metabolites, as well as specific targeted metabolites, in live and fixed cells. Raman imaging and spectral analysis essentially serves as a multiplex functional assay for metabolites that rapidly respond to environmental stimuli, and so provides a powerful complement to mass spectrometry and fluorescence detection methods. We showed the value of metabolic analysis by imaging a series of patient-derived, BRAF mutant melanoma cell lines, representing different de-differentiation phenotypes. The subcellular metabolic heterogeneity across these cell lines is effectively captured by Raman and used to mine for phenotype-dependent, druggable metabolic susceptibilities. We termed this approach as subcellular pharmaco-metabolomics.

In many cancers, mesenchymal-like cells exhibit invasive characteristics, as well as innate drug-resistance to targeted or even immune-therapies[32,65–67]. We hypothesized that the maintenance of such characteristics required lipid biochemical processes that could be mined for druggable susceptibilities. To this end, we utilized a comparative analysis of Raman spectro-imaging on intracellular LDs to identify that lipid-unsaturation associated metabolic activities were uniquely upregulated in the mesenchymal M381 phenotype, as depicted in Fig. 7. This picture is supported by several findings. First, from SRS imaging of deuterated fatty acids and glucose, M381 cells exhibited the lowest relative activities for both lipid uptake (e.g., OA, 18:1) and de novo fatty acid synthesis. Such low metabolic activity might contribute toward making M381 cells insensitive to BRAFi[68]. Second, incubation with SCD1 inhibitor, CAY, which blocks the conversion of SFA to MUFA, led to an imbalance of intracellular SFA and UFA. This imbalance drives the release of UFA stored in M381 LDs to restore the balance. This suggests an intracellular UFA reservoir function for these droplets. Prolonged SCD1 inhibition eventually depletes these LDs of UFA, leading to an excess of SFA in M381. This excess, in turn, contributes toward a type of lipotoxicity through the formation of a phase-separated SM domain. The accompanying loss of membrane fluidity and exclusion of membrane-residing proteins are then associated with an induced apoptosis—a cell fate that can be avoided by

supplying extra MUFA in the culture medium. The susceptibility of SCD1 is uniquely revealed by subcellular Raman analysis, but is not reflected in the bulk transcriptomics (Supplementary Fig. 8c) or bulk metabolomics (Fig. 3h). Both the mechanism and applicability underlying reported susceptibilities in our work are distinctly different from previous reports that mainly relied on bulk analysis[69–71]. This demonstration thus emphasizes the unique value of subcellular pharmaco-metabolomics as a revelatory tool for uncovering new cell biology.

The work here provides an important proof of concept for the use of Raman spectro-microscopy in identifying phenotype-dependent metabolic susceptibilities in cancer cells. It is likely that we are just beginning to mine for how different metabolites are processed and utilized within different cellular sub-compartments. Our current subcellular investigations focus on the spectral region of 2800–3100 cm$^{-1}$, but can be readily extended to additional windows within the fingerprint spectral region to permit the identification of additional metabolite classes[19–21]. Other subcellular structures could be probed similarly to how the LDs were analyzed here, to resolve a more comprehensive intracellular picture of the organelle network, such as the membrane-bound organelles of ER and the Golgi apparatus[46]. Another aspect that worth exploration is the generality of the cell line specific results reported here. For example, whether the susceptibility of SCD1, as revealed in the mesenchymal M381 cells, applies more generally across mesenchymal BRAF mutant melanoma tumors, is both intriguing and important, given the challenges in drugging such tumors. A second challenge will be to extend these Raman tools, in conjunction with surprisal analysis, to characterize the metabolic heterogeneity within intact tissues, and more physiologically relevant environments[17]. Such studies will further validate the general applicability of specific targets identified here and perhaps open up avenues for clinical translation.

## Methods

**Cell lines, chemicals, and cell culture.** Patient-derived melanoma cell lines used in this study were generated from patient's biopsies with informed consent from all subjects under UCLA IRB approval # 11–003254. Cells were cultured in RPMI 1640 (Gibco, 11875119), supplemented with 10% fetal bovine serum (Omega, FB-12), and 0.2% MycoZap Plus-CL antibiotics (Lonza, VZA-2011). Cultures were incubated in a water-saturated incubator at 37 °C with 5% CO₂. Cells were maintained and tested for mycoplasma using kit (Lonza, LT07-118).

Cerulenin (Sigma, C2389-5MG), TVB 3166 (Sigma, SML1694-5MG), CAY10566 (Cayman, 10012562), SC 26196 (Cayman, 10792) were dissolved in DMSO (ATCC, 4-X) at designated concentrations before adding to cell culture media. To conduct cell viability assay, 30k to 50k cells were seeded into six well dishes (Corning, 3516). After culturing for 2 days, growth medium was replaced with fresh medium containing drugs with indicated concentration, and the incubation continues for another 3 days. Cell viability was measured by counting cell numbers of each well with trypan blue. Cell number in vehicle (with DMSO as vehicle) well was used as normalization.

**Stimulated Raman scattering microscopy.** The configuration is shown in Supplementary Fig. 5a. An integrated laser (picoEMERALD, Applied Physics and Electronics, Inc.) is used as a light source for both pump and stokes beams. It produces 2 ps pump (tunable from 770 to 990 nm, bandwidth 0.5 nm, spectral bandwidth ~7 cm$^{-1}$) and stokes (1031.2 nm, spectral bandwidth 10 cm$^{-1}$) beams with 80 MHz repetition rate. Stokes beam is modulated at 20 MHz by an internal electro-optic modulator. The spatially and temporally overlapped pump and stokes beams are introduced into an inverted multiphoton laser scanning microscopy (FV3000, Olympus), and then focused onto the sample by a 25× water objective (XLPLN25XWMP, 1.05 N.A., Olympus). Transmitted pump and stokes beams are collected by a high N.A. condenser lens (oil immersion, 1.4 N.A., Olympus) and pass through a bandpass filter (893/209 BrightLine, 25 mm, AVR Optics) to filter out stokes beam. A large area (10 × 10 mm) Si photodiode (S3590-09, Hamamatsu) is used to measure the remaining pump beam intensity. 64 V DC voltage is used on the photodiode to increase saturation threshold and reduce response time. The output current is terminated by a 50 Ω terminator and pre-filtered by an 19.2–23.6-MHz band-pass filter (BBP-21.4+, Mini-Circuits) to reduce laser and scanning noise. The signal is then demodulated by a lock-in amplifier (SR844, Stanford Research Systems) at the modulation frequency. The in-phase X output is fed back

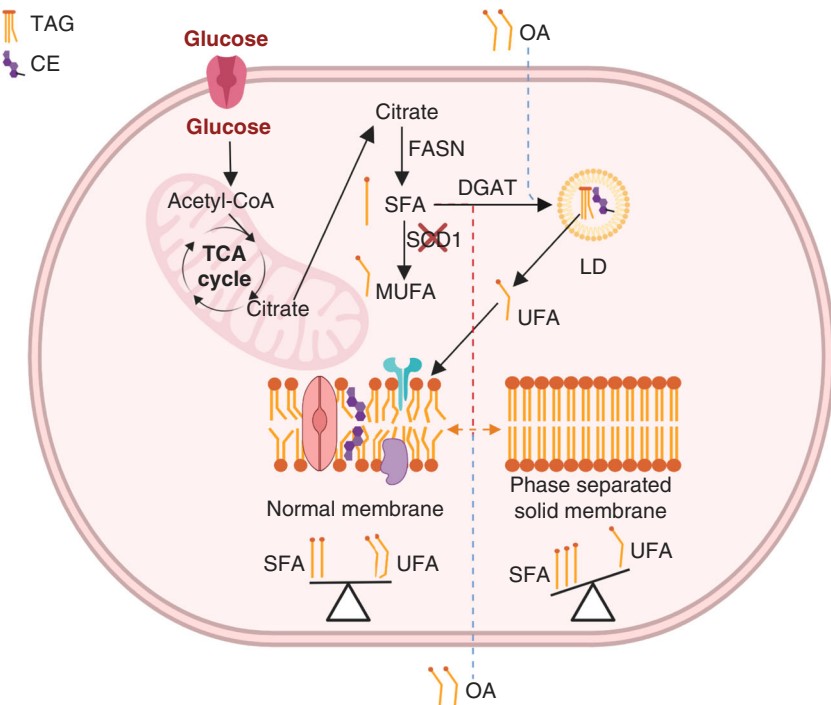

**Fig. 7 Schematic of the proposed cellular metabolic processes for M381 cells under SCD1 inhibition.** SCD1 inhibition blocks de novo MUFA synthesis from SFA, which leads to an imbalance of intracellular SFA and UFA. This imbalance drives the release of UFAs stored in M381 lipid droplets, which act as reservoirs of unsaturated lipids, to restore the balance between SFA and UFA. Prolonged SCD1 inhibition eventually depletes the stored UFA. The resulting imbalance between SFA and UFA transforms fluid normal membrane domains into phase-separated solid membranes. The accompanied loss of membrane fluidity and exclusion of membrane-residing proteins are associated with an induced apoptosis—a cell fate that can be rescued by supplying excess UFA in the culture medium.

to the Olympus IO interface box (FV30-ANALOG) of the microscope. Image acquisition speed is limited by 30 µs time constant set for the lock-in amplifier. Correspondingly, we use 80 µs pixel dwell time, which gives a speed of 8.5 s frame$^{-1}$ for a 320-by-320-pixel field of view. For 2150 cm$^{-1}$ (C-D, d$_7$-glucose), 2109 cm$^{-1}$ (C-D, d$_{31}$-palmitic acid, d$_{33}$-oleic acid, d$_{35}$-stearic acid), 2845 cm$^{-1}$ (CH$_2$) and 2940 cm$^{-1}$ (CH$_3$), the wavelengths of pump laser are 844.1, 847.0, 797.3, and 791.3 nm, respectively. Laser powers are monitored through image acquisition by an internal power meter and power fluctuation are controlled within 5% by the laser system. 16-bit gray scale images are acquired by Fluoview software. SRS spectra were acquired by fixing the stokes beam at 1031.2 nm and scanning the pump beam through the designated wavelength range point by point. SRS spectra were processed and presented by Excel and GraphPad. Lipid-channel (C-H) image was processed from a linear combination algorithm of 5 · [CH$_2$]−0.4 · [CH$_3$] from the CH$_2$ and CH$_3$ images[58]. We recommend that at least three biological replicates with at least five cells in each replicate are acquired for analysis.

**Spontaneous Raman spectroscopy.** Fixed cell pellets were washed two times with pure water, and then resuspended into water to be cell solution to avoid influence from salt crystals after drying. The cell solution containing 5k cells was added dropwisely on a glass slide. After air dry, glass slides with cells were then used to take Raman spectra. Spontaneous Raman spectra were acquired using an upright confocal Raman spectrometer (Horiba Raman microscope; Xplora plus). A 532 nm YAG laser is used to illuminate the sample with a power of 12 mW on sample through a 100×, N.A. 0.9 objective (MPLAN N; Olympus) with 100 µm slit and 500 µm hole. Spectro/Raman shift center was set to be 2000.04 cm$^{-1}$. With a 1200 grating (750 nm), Raman shift ranges from 690.81 to 3141.49 cm$^{-1}$ was acquired to cover whole cellular Raman peaks. Acquisition time for one spectrum was set to be 25 s (5 s times five averaging). The target cell was chosen randomly and spectra of five points (center, top, bottom, left, right) on individual cell were acquired. The acquired spectra were processed by the LabSpec 6 software for baseline correction. Spontaneous Raman spectra were organized and presented by Excel and GraphPad, respectively. To reduce spectral variance for spontaneous Raman spectra caused by intracellular heterogeneity, we recommend that at least three biological replicates with at least ten cells in each replicate are acquired for analysis.

**Coating of imaging dish.** Imaging dish (MatTEK, P35G-1.5-14-C) was coated with 2% sterile gelatin solution (Sigma, G1393) for 30 min, then the coating solution was removed and the dish was left for air dry for another 30 min before using.

**Metabolic deuterium labeling.** Deuterated glucose RPMI 1640 medium was made by supplying d$_7$-glucose (Cambridge Isotope Laboratories, DLM-2062-1) into glucose deficient RPMI 1640 medium (Gibco, 11879020), then completed with 10% fetal bovine serum (Omega, FB-12), and 0.2% MycoZap Plus-CL antibiotics (Lonza, VZA-2011). d$_{31}$-Palmitic acid (Cambridge Isotope Laboratories, DLM-215), d$_{35}$-stearic acid (Cambridge Isotope Laboratories, DLM-379), and d$_{33}$-oleic acid (Cambridge Isotope Laboratories, DLM-1891) were coupled to bovine serum albumin (Sigma, A9418) in 2:1 molar ratio and added to RPMI 1640 complete medium to designated concentration. The resulting solutions was sterile filtered by 0.22 µm low protein binding filter system. Cultured melanoma cells were seeded onto an imaging dish to optimal confluency. The cells were then grown in the corresponding deuterated medium (e.g., 11.1 mM d-glucose used in Fig. 2f, 50 µM d$_{31}$-palmitic acid and d$_{35}$-stearic acid used in Supplementary Fig. 9a, c) for 3 days before fixation and imaging.

**Ratio image processing and data analysis.** Images are analyzed and assigned color by ImageJ. For CH$_2$/CH$_3$ ratio imaging, a threshold (mask) image was first generated by adjusting threshold using Huang method, then nonzero values were normalized to one. CH$_2$ images were then divided by the same set of CH$_3$, and the resulting ratio image multiplied with mask image to create the final CH$_2$/CH$_3$ ratio image. The display range of CH$_2$/CH$_3$ ratio images is set to be 0 - 0.5.

**Fatty acid analysis.** Five million cells were harvested, frozen, and lyophilized overnight. Fatty acid methyl esters (FAMEs) were produced from biomass in a combined extraction, hydrolysis, and derivatization procedure based on previous methods[72]. For each sample, dried biomass was mixed with 2 ml of methylation mixture (20:1 v/v anhydrous methanol/acetyl chloride) and 1 ml hexane and reacted in sealed VOA vials at 100 °C for 10 min. After cooling, 2 mL deionized water was added to the mixture followed by three times extraction with 2 ml hexane. The hexane solution was then treated with anhydrous Na$_2$SO$_4$ to remove residual water and concentrated under a steam of N$_2$ to a final volume of 0.5 ml. FAMEs were identified via gas chromatography/mass spectrometry (GC/MS) on a Thermo Fisher Scientific ISQ by injecting 1 µl of sample in splitless mode. Chromatographic separation was achieved on a ZB-5ms capillary column (30 m by 0.25 mm; film thickness, 0.25 µm). Peaks were identified by comparing the mass spectra and retention times to the authentic standards and library data. Quantification was achieved by a flame ionization detector. To avoid complications from sample loss at sample preparation stage, we used the relative abundance of each species of fatty acids for data interpretation. Relative abundances were calculated by dividing the

peak area for each of the six most abundant fatty acids (16:0, 16:1, 18:0, 18:1, 18:2, and 20:4) to the sum of peak areas of all six species. Data were processed by Excel. The signals of other species are too low and mostly buried in noise.

**Detergent wash**. PBS solution containing 0.5% Triton X-100 (Sigma, T8787), short as PBS-T solution, was used to wash cells in imaging dish[58]. Gently add 1 ml PBS-T detergent solution (above) into imaging dish and place the dish in 4 °C for 10 min. Then the PBS-T washing solution was gently removed and the samples were washed with PBS for two times before imaging.

**RNA extraction, library construction, and sequencing**. Total RNA was extracted from frozen cells pellets (~1 million cells) using the RNeasy Micro Kit (Qiagen, 74004) according to the manufacturer's protocol. Then the RNA sequencing (RNA-seq) was performed using BGISEQ-500 platform at BGI Genomics (Wuhan, China). The library preparation was followed by BGI's standard procedure.

**RNA-seq data dimension reduction and clustering analysis**. Sequencing reads were mapped and aligned to Human Reference Genome (UCSC hg 19) with TopHat. Assembled transcripts for each sample were generated from mapped reads using Cufflinks. All assemblies were combined into a single assembly by Cuff-compare for differential expression analysis. Expression levels in fragments per kilobase of exon per million fragments mapped were generated using Cuffdiff as normalized read counts.

Heatmap and clustering analysis of transcriptomic dataset was performed via MATLAB. Hierarchical clustering was performed with average linkage and Euclidean distance metric. Transcriptomic data of 30 BRAF but not NRAS mutated melanoma patient derived cell lines from the Gene Expression Omnibus database (GEO)[27] were chosen for dimension reduction and clustering analysis. Gene expression of the whole transcriptome or metabolic subset (with all metabolic genes defined from reference[73]) were project onto the top two most dominant constraints defined from surprisal analysis[38]. This way, cell lines with similar whole transcriptomic profiles or metabolic-related gene expression profiles were projected nearby to each other. Cell lines were color-coded based on their respective phenotypes. Top 100 cell phenotype-specific metabolic genes for each of the phenotype are selected based on gene's contribution score toward each phenotype as listed in Supplementary Table 1. Contribution score of each gene to each phenotype are calculated based on gene's contribution score toward the $X$-axis ($G1$) and $Y$-axis ($G2$) in the two-dimensional map ($G1$ and $G2$ values from surprisal analysis). Detailed equations are listed as the following: contribution score of melanocytic phenotype, $S_{melanocytic}$ ($S_1$) $= -G1 - G2$; contribution score of transitory phenotype, $S_{transitory}$ ($S_2$) $= -G1 + G2$; contribution score of neural-crest phenotype, $S_{neural-crest}$ ($S_3$) $= G1 + G2$; contribution score of undifferentiated phenotype, $S_{undifferentiated}$ ($S_4$) $= G1 - G2$. Heatmap of all 400 phenotypic-specific metabolic genes are plotted in Supplementary Fig. 1 and heatmap for a few representative phenotype markers and phenotypic-specific metabolic genes are shown in Fig. 1b.

For $CH_2/CH_3$ correlation analysis across five cell lines, spearman correlation was calculated between each gene and the measured $CH_2/CH_3$ ratio across all five cell lines, where genes that displayed the highest positive or negative correlation with $CH_2/CH_3$ ratio (Spearman $> 0.95$ or $< -0.95$) were further mined for their function through enrichment analysis.

Gene set enrichment analysis (GSEA)[74] was performed using GSEA v4.0.1 with 1000 geneset permutations. Normalized enrichment score was assessed across the curated Molecular Signatures Database (MSigDB) Hallmark, C2 curated gene sets, C4 computational gene sets and C5 gene ontology gene sets. To identify biological processes and pathways most correlated with $CH_2/CH_3$ ratio, we first ranked the genes based on the Spearman correlation between their expression and $CH_2/CH_3$ ratio across all five melanoma cell lines and then performed the pre-ranked option of GSEA with 1000 permutations.

**Surprisal analysis of Raman spectra**. Surprisal analysis was applied as previously described[38]. Briefly, the measured Raman peak signal at certain wavenumber $i$ at cell $c$, $\ln Xi(c)$, is expressed as a sum of a steady state term $\ln^0 Xi$ $(c)$, and several constraints (modules) $\lambda j(c) \times Gij$ representing deviations from the steady state. Each deviation term is a product of a cell-dependent weight (influence score) of the constraint $\lambda j(c)$, and the cell-independent contribution of the wavenumber peak to that constraint $Gij$. Peaks $i$ with high positive or negative $Gij$ values are the ones that are positively or negatively correlated with constraint (module) $j$, which can be used to infer the meaning of each module. To implement surprisal analysis, we first utilized singular value decomposition, which factors this matrix $\ln Xi(c)$ in a way that determines the initial estimate of the two sets of parameters that are needed in surprisal analysis: the Lagrange multipliers ($\lambda j$) for all constraints at a given cell, and for all cell the $Gij$ (cell-independent) analyte patterns for all analyte $i$ at each constraint $j$. Further interaction is implemented when necessary to stabilize the steady state and refining the constraints.

**Incucyte cell apoptosis assay**. Cells were seeded and monitored using an Incu-Cyte® S3 live-cell imaging system (Essen BioScience). Cells were exposed to drug treatments for up to 72 h in the presence of IncuCyte® Caspase-3/7 Green apoptosis

dye (Essen BioScience, Cat. No. 4440). Images were taken at 20-min intervals from nine separate regions per well using a 20× objective. Apoptotic cell counts per well at each time point were quantified using the IncuCyte Basic Analyzer.

**Migration and invasion assays**. Transwell chambers coated with (Corning, 354480) and without matrigel (Corning, 354578), respectively were utilized to conduct the invasion and migration assays according to manufacturer's protocol. Briefly, cells received indicated treatments three days before the assays. At the start of the assays, cells were harvested and counted, and 50k ml$^{-1}$ cells suspension was prepared. 0.5 ml of cell suspension was added to the upper chamber of the 24-well chambers. The media in lower chamber contains 10% FBS. Cells were allowed to migrate for 22 h at 37 °C. The transwell membranes were then fixed and stained with 0.05% crystal violet solution. A cotton swab was used to remove cells that had not migrated or invaded through the chamber. Then, a fluorescence microscope was used to image the migrated or invaded cells, and four fields were independently counted from each migration or invasion chamber. Two or four biological replicates of experiments were conducted.

**Generation of SCD1 stably knockdown cells**. M381 cells were transfected with shRNA lentiviral particles targeting SCD1 (Santa Cruz, sc-36464-V) following the manufacturer's protocol. Scrambled shRNA lentiviral particles (Santa Cruz, sc-108080) were used as a control. Stably transfected cells were selected with 1 μg ml$^{-1}$ puromycin (Thermo Fisher, A1113803).

**Lipidomics profiling**. Cells going through indicated treatments were harvested as frozen pellets. Lipids were extracted using methyl tert-butyl ether (MTBE)/ methanol after the addition of 54 isotope labeled internal standards across 13 lipid classes. The extracts were concentrated under nitrogen and reconstituted in 10 mM ammonium acetate in dichloromethan:methanol (50:50). Lipids were analyzed using the Sciex Lipidyzer platform consisting of a Shimadzu LC and AB Sciex QTRAP 5500 LC-MS/MS system equipped with SelexION for differential mobility spectrometry (DMS). Multiple reaction monitoring (MRM) was used to target and quantify over 1000 lipids in positive and negative ionization modes with and without DMS. The resulting lipidomics data are provided as Supplementary Data 1.

**Reporting summary**. Further information on research design is available in the Nature Research Reporting Summary linked to this article.

## Data availability
All the data supporting the findings of this study are available within the article and its Supplementary Information files and from the corresponding author upon reasonable request. The following databases are used: Human Reference Genome (UCSC hg 19), Gene Expression Omnibus database (GEO), Molecular Signatures Database (MSigDB). RNA-seq data have been deposited to array express with accession number of E-MTAB-8842. Source data are provided with this paper.

## Code availability
Custom code for the surprisal analysis of Raman spectra has previously been published and deposited on GitHub (https://github.com/mesako/Melanoma-Publication) [36]. Source data are provided with this paper.

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

## Acknowledgements

L.W. acknowledges start-up fund support from California Institute of Technology. We acknowledge the following agencies and foundations for support: NIH Grant U01 CA217655 (to J.R.H.); the Parker Institute for Cancer Immunotherapy (J.R.H. and A.R.), the WA State Andy Hill CARE Foundation (J.R.H.), and an ISB Innovator Award (Y.S.).

## Author contributions

L.W., J.R.H., J.D., and Y.S. conceived the study and designed the experiments. J.D., Y.S., C.Q., D.Y., K.M., D.L., A.N., and R.W. performed the experiments. J.D. and Y.S. analyzed and interpreted the data. L.W., J.R.H., A.R., and R.L. provided conceptual advice on data analysis and interpretation. L.W., J.D., J.R.H., and Y.S. wrote the manuscript. L.W. and J.R.H. supervised this study.

## Competing interests

The authors declare no competing interests.
