## [Peer Review File · Nature Communications]

Reviewers' comments:

Reviewer #1 (Remarks to the Author):

This paper seeks to justify the use of stimulated Raman (SRS) imaging to explore what the authors term the pharmaco-metabolomics of a set of BRAF-1 mutant cell lines. An enormous amount of data is presented in support of the hypothesis that Raman microspectroscopy can be used to elucidate elements of metabolomics at single cell scale. Overall, this is a carefully executed experimental study, and the paper is well-written. I think it should ultimately be published. There were, however, a few issues that require further explanation and/or justification.

The first major issue is the oversimplified assignment of the complex SRS line shape in the C-H stretching region, specifically assigning an unresolved feature at 2845 cm^{-1} to lipids and a similar feature at 2940 cm^{-1} to protein. I am certain that the authors appreciate the approximate nature of these statements and assignments, but in order to orient the more general reader, they should supply explicit justification for these assignments, especially for the spectral data that was not obtained explicitly from lipid droplets. While it may be justified to think of these bands as being dominated by the specified molecular constituents, a more nuanced discussion of the contribution of individual cellular components to the complex line shape is warranted.

I also find it somewhat surprising that the authors went to a lot of trouble to perform careful surprisal analysis (SA), yielding a set of decreasing only informative constraint values but then did not use this information to map spectral response. Notwithstanding the issues identified in the paragraph above, a more informative picture of the differences between cell lines and treatment conditions would be obtained by mapping constraint values as heat maps rather than presenting Raman images traced back to a single band, especially in those instances where that band is poorly resolved as in the case of the 2845 and 2940 cm^{-1} bands.

Finally, the authors go to great lengths to justify the interpretation of the observed trends in the Raman spectral data, and there are some very nice correlations between things like "CH₂/CH₃" intensity ratios and expression levels. While this point may seem minor, it would be interesting if the authors would comment on the fruitfulness of a hypothesis driven investigation of the Raman imaging data as opposed to a more discovery oriented approach in which a reasonable set of spectral indicators are cross correlated against all possible measures of gene expression. One gets the impression that statistically significant correlations can always be found in such a large, rich, and interesting data set.

In addition to these general issues, there were some minor points that need to be addressed and revision.

Figure 1(e) and 1(f):

With respect to constraint λ_1 there is an enormous amount of variability in the data. Some cell lines, e.g. M262, have very blue (negative) colors, while other cell lines, e.g. M381, exhibit very red (positive) color, and the error bars in 1(f) are huge. For example, in M262: value is ~ 210 a.u., while error bar ~ 190 a.u., etc. Can the authors comment on the magnitudes of the variations in λ_1 values even within a single cell line.

Figure S5(f):

In the SRS images, after adding OA in the 60 h CAY-treated cells, set 1 has a huge difference in CH2/CH3 but not set 2. Is 10 hour a required time duration for the treatment? Would the equilibrium be break again if the cells are putting under this set-up for longer period? The authors didn't explain why they chose the 60 h or 10 h time windows.

There are a few minor errors, e.g. referencing the wrong figure panel of Figs. S1c/S1d and Figs. 1f/1g on p. 4.

Reviewer #2 (Remarks to the Author):

The manuscript by Du et al., attempts to provide three distinct highlights, one the use of Raman spectro-microscopy to probe metabolites in living single cells, second to show that degree of de-differentiation increases with progression to melanoma cells (also described in the Cell Cancer paper published by the authors previously) and third that fatty acid pathways offer druggable targets in de-differentiated mesenchymal cell lines. While the use of Raman spectro-microscopy for single cell metabolomics is noteworthy, the conclusions drawn in the manuscript are not justified and warrant significant control experiments. This reviewer describes the major issues with the manuscript below. A major revision addressing these issues would be worthy of further consideration.

In order to identify metabolomic signatures of de-differentiation, the authors use a combination of transcriptomic and metabolomic analysis. The authors use transcriptomic data from 31 of 53

patient-derived BRAF-mutant melanoma cell lines (those that do not have RAS mutations). This is a reasonable choice. In their Cell Cancer paper (reference 27 as cited in the manuscript), the authors used a PCA method to cluster the 53 cell lines and obtained a pseudo separation with processes that were mainly those involved in de-differentiation). This PCA is completely different from what is presented in Figure 1a. How did they get a completely different result? Additionally, also in Figure 1a, the authors present a PCA plot considering only 1600 metabolic genes. This PCA plot is also so clearly delineated to defy normal analysis. No description of methods is provided anywhere which would show validation of these results (in addition there are 28 and 26 points respectively in the PCA plots). In Figure 1b, they present a heat map of cherry-picked genes from the 1600 genes. How was this choice made? Even if it is assumed that the authors consider only Fatty acid related, Amino-acid related, nucleotide-related and glycosylation-related genes, these would add up to several hundreds and not to 22 genes. The heat map provide a fundamental concern on the conclusions reached. Why not present a heat map of all the hundreds of genes involved in these processes, or even simply present a heat map of all fatty acid-related genes instead of using 7 genes. Figures 1 a and b are not credible and appear to be chosen to fit a hypothesis.

The idea of using Raman Spectra and the surprisal analysis are both exciting aspects of the manuscript. The authors have chosen 5 representative cell lines. What is the variance if more cell lines are chosen (this needs to be demonstrated)? How did they choose the 5 cell lines? How was the averaging done for each of the 10 cells/line? Assuming the variance is low and the deconvolution is reasonable, the authors arrive at CH₃ and CH₂ vibrations and see the differences between these across the cell line types. They conclude that CH₂/CH₃ ratio decreases with the progression of dedifferentiation. If the analysis is correct, this could even be a true observation. However, the conclusions they draw from this are not warranted without a significant number of control experiments. The authors conclude that elevated FASN activity in the more differentiated melanoma cell lines contributes to metabolic susceptibility. This conclusion was arrived based on the CH₂/CH₃ ratio. Alternate hypotheses would also fit such a result as described below. In the process of de-differentiation, there are significant histone modifications, involving demethylation, methylation, etc. Why would this not contribute to the CH₂/CH₃ ratio? An experiment like a genome wide ChIP-seq with H3K27/H3K27Ac and H3K4 could help in identifying the degree of methylation changes. The authors focus on fatty acid synthesis. How about alterations in other lipids? Did they examine the 2 major lipid processing pathways, one involving acyl CoA processes (considered by the authors) and the other involving acetyl CoA (not analyzed by the authors) involving cholesterol and sphingolipid pathways? Further the authors in the cell prep use FCS which contains fatty acids including arachidonic acid which is processed in the cells to produce fatty acyl derivatives (in fact the increase in Phospholipase gene levels may point to this processing). What controls were used to eliminate this confounding effect?

The authors describe that M381 phenotype accumulates more unsaturated lipids and cholesteryl esters in lipid droplets. They describe their prior work on lipid peroxidation in mesenchymal-specific GPX4-inhibitor induced ferroptosis. What is the cross talk between oxidative processes and lipid saturation. The GC-MS results are shown without details. What standards were used in

quantification of the fatty acids. Figure 3 states that percentages of 16:0, 16:1, 18:0, 18:1, 18:2, and 20:4 are normalized to all extracted fatty acids (n=4-5). More details would help in understanding how the relative quantification was achieved.

In discussing lipid desaturases, the authors point to their involvement in elevated lipid-droplet unsaturation of the M381 mesenchyme phenotype. Did the authors control for PUFA in the media? How were quantifications achieved? In Figure 4d were relative viability studies carried out for longer time to assess if lines other than M381 also showed apoptotic effects? This is important to the conclusion that CAY inhibition of SCD1 for MUFA synthesis leads to more significant viability loss for M381 cell lines.

While the broad conclusions are interesting and perhaps can be useful for drug targeting, their validity should be demonstrated through more rigorous control experiments.

Reviewer #3 (Remarks to the Author):

The manuscript describes the use of Raman spectroscopic imaging to monitor changes in lipid metabolism associated with cellular differentiation and progression of melanoma. The results contain a combination of Raman images and corresponding transcriptomics and lipidomics. Results show changes in vibrational bands detected in images of melanoma cell lines with well-known and studied phenotypes. Features detected in Raman images point to changes in lipid saturation that correlate with mechanisms implicated by the transcriptomic data. The significance is a technique that can monitor metabolic changes in single live cells

The manuscript suggests that Raman can be broadly applied to monitor metabolic changes in cells. The use of Raman scattering to detect lipid droplets is well established as cited in the manuscript (e.g. Ref. 22). In particular, the high concentration of lipids confined in lipid droplets are readily detectable in Raman experiments. Can the authors comment on quantitative metrics associated with metabolic changes? For example, what concentration is necessary for detection? It is not clear this methodology can be widely applied beyond the CH vibrations detected with lipids or with isotope tags (i.e. the deuterium incorporation through D-labeled glucose metabolism) that exist at otherwise vibrationally quiet frequencies.

It appears the cell results reported are determined from fixed cells. This has real significance for live cell studies. What is the potential for live cell imaging? How does the variance from individual cells compare to the trends observed in the averaged data presented? How many cells are necessary in such experiments?

Overall this is an impressive study. The detection of drug susceptibility of the SCD1 pathway in the Raman data, not detected in transcriptomic or through other metabolomic techniques, validates a new approach to studying metabolic changes relevant to cancer.

Reviewer #4 (Remarks to the Author):

Based on Raman spectromicroscopy and SRS to subcellularly monitor metabolites in a series of patient-derived BRAF mutant melanoma cell lines the authors have identified the fatty acid synthesis pathway and lipid mono-unsaturation, respectively, as druggable susceptibilities for differentiated and de-differentiated melanoma cell lines. This paper provides an interesting proof of concept for the use of Raman spectro-microscopy in identifying phenotype-dependent metabolites in living cells at the single cell level. However, in the current setting of cell cultures, the full potential and benefit of this method has not been shown and the identified targets may not be as novel as the authors claim.

Specific comments:

1. The authors provide a nice proof of principle of the use of Raman technologies for probing metabolites at the single cell level. Unfortunately, restricting to rather uniform cell line cultures, the authors fail to show the full potential and benefit of these approaches for single cell analysis. I would argue that the targets that have been identified could also have been identified using standard lipidomics approaches on bulk. Inclusion of a more heterogeneous model system would have substantially augmented the merit of this paper.
2. Along the same lines, the concepts and targets that have been identified may not be as novel as the authors claim. De novo lipogenesis have been extensively documented as potential target in melanoma (Talebi et al., 2018; Wu et al., 2018;...). Also, SCD1 is a known target in studies on melanoma (e.g. Pisanu et al, 2018). The involvement of detergent-resistant microdomains and lipid droplets as a possible reservoir for unsaturated fatty acids have also been reported.

3. Cerulenin cannot be used as a bona fide inhibitor of fatty acid synthesis. Cerulenin is notoriously promiscuous and affects several other targets. Nowadays, more selective inhibitors such as TVB-3166 or IPI-9119 are available.

4. In Fig 1c, the intensity scale in the Y axis is confusing as the spectra are offset apart for presentation.

Response to Reviewer #1

We thank the reviewer for the constructive and insightful comments that would help make our paper clearer and more organized. We particularly appreciate the reviewer's recognition that "*an enormous amount of data is presented in support of the hypothesis that Raman microspectroscopy can be used to elucidate elements of metabolomics at single cell scale*". We also thank the reviewer for the encouraging comments that "*Overall, this is a carefully executed experimental study, and the paper is well-written. I think it should ultimately be published*".

We understand that the reviewer has some concern on the data justifications and explanations. Further explanations would indeed help better clarify our points and make our paper stronger. We believe that we have addressed these questions in our revised manuscript with new data and analysis. Below, we would like to give a point-by-point response.

Reviewer #1, Comment 1. *The first major issue is the oversimplified assignment of the complex SRS line shape in the C-H stretching region, specifically assigning an unresolved feature at 2845 cm⁻¹ to lipids and a similar feature at 2940 cm⁻¹ to protein. I am certain that the authors appreciate the approximate nature of these statements and assignments, but in order to orient the more general reader, they should supply explicit justification for these assignments, especially for the spectral data that was not obtained explicitly from lipid droplets. While it may be justified to think of these bands as being dominated by the specified molecular constituents, a more nuanced discussion of the contribution of individual cellular components to the complex line shape is warranted.*

Response: We appreciate the reviewer's comment on the need of more in-depth explanations for Raman spectral assignments. The Raman peak positions and spectral shapes are specific to chemical bonds (e.g. C-H, C-C, C-N, C-O, P-O, C=O) and their associated vibrational motions (e.g. symmetric, antisymmetric stretching). The large Raman peaks in the range of 2800-3100 cm⁻¹ from cells (**Fig. 1c**) are assigned to C-H bonds, the most abundant bonds in biological samples. The peaks at 2845 cm⁻¹ and 2940 cm⁻¹ are assigned to the symmetric stretching motion of CH₂ and CH₃, respectively. While CH₂ and CH₃ are shared by many endogenous biomolecules, lipids and proteins are the dominant sources of CH₂ and CH₃ structures in cells, due to both their high abundance in the cells (> mM concentration) and their specific chemical structures rich in CH₂ and CH₃, respectively. The long fatty acyl chains of lipids are composed of CH₂ structures. In addition, each protein has on average of ~ 438 amino acids¹, which are abundant with CH₃ structures in their side chains. This is why it is well documented in the Raman literature that 2845 cm⁻¹ and 2940 cm⁻¹ bands designate overall lipids and proteins in biological samples, respectively²⁻⁶.

To better illustrate our assignments, we first performed Raman spectra measurements on pure proteins (bovine serum albumin, BSA) and lipids (1,2-dioleoyl phosphocholine, PC) as well as from the cytoplasm and nucleus of cells from one selected cell line, the M381 cells. In addition, we obtained molecule-specific Raman spectral information by comparing spectra directly from M381 cells before and after protease k wash (i.e. deproteinated) and triton wash (i.e. delipidated), respectively (**Fig. R1**). Our data collectively support the assignments of 2845 cm⁻¹ and 2940 cm⁻¹ peaks to lipids and proteins with following evidence. First, **Fig. R1a** confirms that 2845 cm⁻¹ and 2940 cm⁻¹ are the characteristic peaks of lipids and proteins. Second, it is widely known that cytoplasm has higher lipid content than nucleus. This is consistent with the higher intensity of 2845 cm⁻¹ peak from cytoplasm in **Fig. R1b**. Third, we adopted a protein-specific washing experiment with protease k (ProK), which extensively digests proteins in cells. As shown in **Fig. R1c**, the 2940 cm⁻¹ peak from M381 cells significantly decreased after ProK wash. Fourth, our

lipid-specific washing experiment with Triton, which washes away lipids, selectively lowered the 2845 cm^{-1} peak in **Fig. R1d**. These four evidences hence support our assignments of 2845 and 2940 cm^{-1} peaks to cellular lipids and proteins. We have now added the **Fig. R1** as our new **Fig. S4** along with explanations for the C-H peak assignment in the caption.

Fig. R1 Spontaneous Raman spectra characterizations for lipids and proteins *in vitro* and in cells. a) Raman spectra of bovine serum albumin (BSA), representing the general protein structures (blue), and 1,2-dioleoyl phosphocholine (PC, the highest abundant lipid in M381 cells from lipidomics), representing the overall lipid structures (red). 2845 cm^{-1} and 2940 cm^{-1} peaks are dash-line highlighted (the same for all sub-figures). b) Raman spectra from cytoplasm (red) and nucleus (blue) of M381 cells. c) Raman spectra of M381 cells before (CT, control, black) and after protease k (ProK, red) treatment. d) Raman spectra of M381 cells before (CT, black) and after triton (Triton, red) treatment. Spectra in a) and b) are self-normalized, in c) and d) are normalized to 2845 cm^{-1} and 2940 cm^{-1} , respectively.

Reviewer #1, Comment 2. *I also find it somewhat surprising that the authors went to a lot of trouble to perform careful surprisal analysis (SA), yielding a set of decreasing only informative constraint values but then did not use this information to map spectral response. Notwithstanding the issues identified in the paragraph above, a more informative picture of the differences between cell lines and treatment conditions would be obtained by mapping constraints values as heat maps rather than presenting Raman images traced back to a single band, especially in those instances where that band is poorly resolved as in the case of the 2845 and 2940 cm^{-1} bands.*

Response: We thank the reviewer for this comment and for the recognition of our informative constraints from surprisal analysis. The reviewer suggests that it may be more informative if we map out spectral response between cell lines and treatment through surprisal analysis. In light of

the reviewer's suggestion, we explored surprisal analysis for unsupervised analysis our hSRS spectra of lipid droplets in M381 cells before and after treatment from both fatty acid desaturase inhibitors, CAY (SCD1 inhibitor) and SC (delta 6 desaturase inhibitor), with varying doses (i.e. M381 control (CT), 1 μM CAY, 5 μM CAY, 10 μM CAY, 1 μM SC, 5 μM SC). As shown in the heatmap for the constraint values below in **Fig. R2**, surprisal analysis is indeed able to catch the spectral responses to drug treatment. The heatmap from single lipid droplets (**Fig. R2a**) clearly reveals that the score of constraint 1 (λ_1) largely decreases after CAY treatment and is only slightly lowered with SC treatment. Meanwhile, this decrease of λ_1 becomes larger with higher concentration CAY treatment. Such trend can also be quantitatively visualized by plotting of average scores of λ_1 of lipid droplets under different treatment (**Fig. R2b**).

The assignment of the Raman peaks in λ_1 (bottom of **Fig. R2c**) shows the spectral features of total CH_2 (2852 cm^{-1}), cholesteryl esters (band of 2957-2997 cm^{-1}) and unsaturated lipids (3022 cm^{-1}). The combination of the scores from λ_1 (**Fig. R2a-b**) and the spectral features (**Fig. R2c**) implies that both unsaturated lipids and cholesteryl esters decrease in lipid droplets after applying CAY to M381 cells, accompanied by possible accumulation of triacylglycerides (TAG). The accumulation of TAG with SCD1 inhibition in M381 cells may arise from the shuttle of excess saturated fatty acids to lipid droplets, likely for protection against saturated fatty acid cytotoxicity^{7,8}. Indeed, surprisal analysis generates an unbiased and informative picture of the spectral response to treatments, which is consistent with our normalized spectral data in **Fig. 4b-4c**. **Fig. R2** has been added as our new **Fig. S10**. We also added one sentence in the text (page 6): "This spectral response for decreased unsaturation upon drug treatment was also well-reflected in the heatmap of constraint scores by surprisal analysis (Fig. S10)."

We fully appreciate the reviewer's suggestion and expect that spectral analysis that goes beyond the micro lipid droplets to cover the whole cell region and in the full spectra window will bring a more informative picture. We plan to comprehensively explore the information through mapping constraints values as our future goal, noted in the discussion section.

Fig. R2 Surprisal analysis of hSRS spectra of single lipid droplets (LDs) in M381 cells before and after varying doses of CAY and SC treatment. a) Heatmap for scores of the top two constraints (constraint 0 (λ_0) – constraint 1 (λ_1)) by surprisal analysis of hSRS spectra on LDs in M381 cells at different treatment conditions, i.e. M381 control (CT), 1 μ M CAY, 5 μ M CAY, 10 μ M CAY, 1 μ M SC, 5 μ M SC. Each column represents an individual LD and each row represents the constraint scores. b) The average score of λ_1 in a) across six treatment conditions. Data shown as mean \pm SEM. c) Raman peak assignments for λ_0 and λ_1 . The 3022 cm^{-1} peak (violet arrow) is assigned to unsaturated lipids (UL) and the pink shadowed range from 2957 cm^{-1} to 2997 cm^{-1} is assigned to cholesteryl esters (CE). 2908 cm^{-1} is the zero point in λ_1 . n=16-24 LDs per condition.

Reviewer #1, Comment 3. *Finally, the authors go to great lengths to justify the interpretation of the observed trends in the Raman spectral data, and there are some very nice correlations between things like “CH₂/CH₃” intensity ratios and expression levels. While this point may seem minor, it would be interesting if the authors would comment on the fruitfulness of a hypothesis driven investigation of the Raman imaging data as opposed to a more discovery oriented approach in which a reasonable set of spectral indicators are cross correlated against all possible measures of gene expression. One gets the impression that statistically significant correlations can always be found in such a large, rich, and interesting data set.*

Response: While the first half of our manuscript is more discovery driven to validate the methodology toward pharmaco-metabolomics with surprisal analysis and through correlations between the “CH₂/CH₃” intensity ratios and gene expression levels, the second half of the manuscript is more hypothesis driven. Based on the reviewer’s suggestion, we have now emphasized this hypothesis-driven aspect of the lipid droplet study in the manuscript. For example, we added the following sentence into the text (page 4): “We thus hypothesized that a deep interrogation of the lipid biochemistries in these cell lines might reveal additional druggable susceptibilities that distinguish the mesenchymal phenotypes. To this end, we studied the role of lipid storage in lipid droplets.”

In addition, we want to briefly comment on the correlation between Raman and expression data. While we agree with the reviewer that *“statistically significant correlations can always be found in such a large, rich, and interesting data set”*, we did go to some efforts to show that the correlations we highlight are not random. It may be relatively easy to find correlations between “CH₂/CH₃” ratios with several random genes. However, our data between the CH₂/CH₃ ratios and the GESA scoring of whole biological meaningful lipid biosynthesis pathway, **Fig. 2d**), show a highly linear correlation ($R > 0.9$) and should not be a mere coincidence and should indicate the real biological correlation.

Reviewer #1, Comment 4. *Figure 1(e) and 1(f): With respect to constraint λ_1 there is an enormous amount of variability in the data. Some cell lines, e.g. M262, have very blue (negative) colors, while other cell lines, e.g. M381, exhibit very red (positive) color, and the error bars in 1(f) are huge. For example, in M262: value is ~210 a.u., while error bar ~190 a.u., etc. Can the authors comment on the magnitudes of the variations in λ_1 values even within a single cell-line.*

Response: The reviewer points out that, while the scores of constraints λ_1 are statistically significant across cell lines, the error bars are large. We want to note that the high variation is not caused by the spectra per se or from artifacts of surprisal analysis, but due to a limitation of the spontaneous Raman spectrometer. To help better explain this issue, we present **Fig. R3** below to illustrate the detailed data acquisition on a representative cell with spontaneous Raman spectrometer. The focal area of our laser beam in this instrument is small (~1 μ m diam) (green points in **Fig. R3**) relative to the size of the cells (~20 μ m diam, **Fig. R3**). Due to the slow

acquisition speed for spontaneous Raman spectra (around 30 s per spectrum), we acquired 5 spectra from 5 different positions across each fixed cell (designated in **Fig. R3**). Those spectra are averaged to yield a single representative spectrum for one cell. A spectrum collected near the nucleus will vary from that from the cytoplasm, similar to what we showed in **Fig. R1b**. Thus, the error bars originate from the structural heterogeneity of the cells at the length scale of the optical probe, rather than instrumental noise. This is why we validated our results from at least three experimental replicates.

That the error arises from intracellular variations can be proven by SRS images (**Fig. 2a**), which map out the chemical distribution across whole cells with high spatial resolution (~ 450 nm). The error bars for CH_2/CH_3 ratios are significantly smaller in SRS images (**Fig. 2b** compared with **Fig. 1f**) and the nucleus-to-cytoplasm variation can be clearly seen in these images.

As a solution to this issue for our future study, we intend to customize our spectrometer. We will use a higher power laser and expand the laser focus to ~ 20 μm that can cover the whole cell so that spontaneous Raman spectra can be collected without lowering the acquisition speed. This will reduce the error bars by removing the intracellular heterogeneity. A similar strategy is recently adopted in other studies⁹. We have added **Fig. R3** as our new **Fig. S2a**. We also added the following sentences in the text (page 3) to explain this issue: “We note here that the relative high variance in the λ_1 originates from the intracellular heterogeneity from relatively low sampling in spontaneous Raman acquisition (Fig. S2a). This issue is largely bypassed in SRS imaging, as shown below with much higher resolution and sampling.” In addition we indicate the optimal replicates/sample size to ensure small variance for statistically significant analysis for spontaneous Raman spectra in SI methods to illustrate this point: “To reduce spectral variance for spontaneous Raman spectra caused by intracellular heterogeneity, we recommend that at least three biological replicates with at least ten cells in each replicate are acquired for analysis.”

5 spots/cell
10 cells/sample
50 spectra/sample

Fig. R3 A representative cell under widefield mode during spontaneous Raman spectra acquisition. Green points indicate the laser focal points on cells. We selected 5 points (center, top, bottom, left, right) on each cell to acquire Raman spectra and averaged them for one cell. We randomly chose 10 cells to represent one cell line.

Reviewer #1, Comment 5. Figure S5(f): In the SRS images, after adding OA in the 60 h CAY-treated cells, set 1 has a huge difference in \$\text{CH}_2/\text{CH}_3\$ but not set 2. Is 10 hour a required time duration for the treatment? Would the equilibrium be break again if the cells are putting under this set-up for longer period? The authors didn't explain why they chose the 60 h or 10 h time windows.

Response: We thank the reviewer for this comment, which would help better clarify our results. The experiment of **Fig. S5f** (now **Fig. S13f**) is a two-color pulse-chase experiment with deuterated

OA (chase) after the M381 cells are pulsed with CAY (the SCD1 inhibitor) for a period. The pulse promotes the formation of the solid membrane structures from M381 cells, which are viewed in the C-H channel (the red-hot image in **Fig. S13f**, for pre-existing lipids). The white-saturated C-H signals indicate the formation of solid membranes with much higher concentration of C-H bonds compared to normal membranes. This is characterized by our previous triton washing experiment in **Fig. 5a**. The chase reveals that OA can have a rescue effect by being incorporated into solid membranes (cyan-hot image in **Fig. S13f**, for newly incorporated OA). Thus, we expect the solid membrane could gradually diminish during the OA incorporation process.

By correlating the C-H and the C-D images, we found that if the structures have high C-D signals, they show less solid membrane, indicated by greatly diminished C-H signal relative to what is seen for solid membrane (i.e. set 2 in **Fig. S13f**, a representative cell is indicated in the orange square). The reduction of solid membrane is also accompanied with an increased number of lipid droplets, shown as small white dots in both the C-H and C-D images. This is because OA leads to dissolution of the solid membrane and is accompanied by storage of the excess saturated lipids in lipid droplets for rescue. The observation is consistent with a recent report which characterizes lipotoxicity⁸. In contrast, if the structures have low C-D signals with less OA incorporation, they remain as solid membrane with characteristic saturated signals in the C-H channel (i.e. set 1 in **Fig. S13f**, a representative cell is indicated in the white square). This indicates minimum OA rescue. We purposely selected two representative images (set 1 vs set 2) to show both scenarios as evidence for the OA rescue effect in **Fig. S13f**. Similar to the formation of solid membrane structures, which is heterogenous across cells, the dissolution is also heterogeneous, since different cells exhibit variable responses and uptake different amount of OA for rescue. This actually underscores the importance of our single cell imaging method.

Our time window was selected to find a time point where two-color correlative images could capture the heterogeneous response of cells for both cellular behaviors stated above. We found that 60 h incubation with 5 μM CAY drug induces clear formation of solid membrane structures in most cells. In addition, a 10-h chase is a proper time length for capturing the heterogeneity in cellular response. If the time is too short (i.e. < 1 h), cells won't uptake sufficient OA to show the rescue effect. If the time is too long (i.e. > 24 h), then all solid membrane has disappeared. In addition to 10 h OA treatment, we also performed 5 h and 15 h chase treatment, both of which showed similar cell patterns as the 10 h chase (indicated by white or orange squares) (**Fig. R4**). Therefore, it is reasonable to conclude that a chase time in the range of 5 h and 15 h is appropriate. We added the explanations for the choice 60 h and 10 h of incubation time in the caption of **Fig. S13**: "For the choice of time window, our selection rule is to find a time point where two-color correlative images could capture the solid-membrane formation and the heterogeneous rescue response of cells with OA incubation for both cellular behaviors stated above."

We reasoned that the equilibrium won't be broken even for a reasonably longer treatment of 20 μM OA chosen for the chase setting. This is because it is known that the cells can shuttle the excess cellular fatty acids to lipid droplets as a self-protection mechanism to maintain fatty acid equilibrium within cells. Additionally, OA has higher propensity than other fatty acids to be channeled into lipids and stored in lipid droplets^{10,11}. As shown in **Fig. S9c**, a longer treatment (72 h) with a much higher doses of OA (50 μM) resulted in an increased number of lipid droplets in all cell lines. However, such treatment did not cause any cytotoxicity according to the viability assay in M381 cells (**Fig. S13a**). This proves the maintenance of intracellular balance for fatty-acid compositions for membrane structures, which is essential for cellular functions.

Fig. R4 Pulse-chase experiment with CAY treatment and OA rescue for varied chase time duration. M381 cells were firstly incubated in medium containing 5 μM CAY for 60 h (pulse). The medium was then changed to fresh medium containing 20 μM $\text{d}_{33}\text{-OA}$ for 5 h (left), 10 h (middle) or 15 h (right) incubation (chase), respectively. Representative C-H lipid channel (red hot) and C-D channel (cyan hot) images for two selected sets of cells were shown for each time window of OA treatment. White-squared cells in set 1 represent cells with low C-D signals from $\text{d}_{33}\text{-OA}$ (cyan hot) and high C-H signals from pre-existing lipids (red hot). Orange-squared cells in set 2 represent cells with high C-D signals from $\text{d}_{33}\text{-OA}$ (cyan hot) and low C-H signals from pre-existing lipids (red hot).

Reviewer #1, Comment 6. There are a few minor errors, e.g. referencing the wrong figure panel of Figs. S1c/S1d and Figs. 1f/1g on p. 4.

Response: We thank the reviewer for the careful checking. These issues have all been corrected.

Response to Reviewer #2

We thank the reviewer for all the careful and thorough reading of our paper and all constructive and insightful comments for helping make our paper clearer. We appreciate the reviewer's recognition of the "three distinct highlights" of our manuscript. In particular, we thank the reviewer for the comments that "*the use of Raman spectro-microscopy for single cell metabolomics is noteworthy*" and "*The idea of using Raman Spectra and the surprisal analysis are both exciting aspects of the manuscript*". We understand that the reviewer had some concerns about the justifications of our conclusions and requested more control experiments. We have performed additional experiments and analyses, which we believe address the reviewer's concerns. In addition, we have revised the paragraphs in the results section based upon the reviewer's comments and have added more detailed experimental description, in particular for RNA-seq data processing, in the SI methods.

Reviewer #2, Comment 1. *In order to identify metabolomic signatures of de-differentiation, the authors use a combination of transcriptomic and metabolomic analysis. The authors use transcriptomic data from 31 of 53 patient-derived BRAF-mutant melanoma cell lines (those that do not have RAS mutations). This is a reasonable choice. In their Cell Cancer paper (reference 27 as cited in the manuscript), the authors used a PCA method to cluster the 53 cell lines and obtained a pseudo separation with processes that were mainly those involved in de-differentiation). This PCA is completely different from what is presented in Figure 1a. How did they get a completely different result? Additionally, also in Figure 1a, the authors present a PCA plot considering only 1600 metabolic genes. This PCA plot is also so clearly delineated to defy normal analysis. No description of methods is provided anywhere which would show validation of these results (in addition there are 28 and 26 points respectively in the PCA plots).*

Response: We apologize for the confusion here. In **Fig. 1a**, we adopted surprisal analysis (SA) instead of PCA to cluster the 30 no-RAS-mutation cell lines selected from the Cancer Cell paper¹² (**Fig. 1C** of the *Cancer Cell* paper, shown in **Fig. R5a** below). SA has been demonstrated as an effective data clustering method for gene expression analysis, yielding similar results to PCA¹³⁻¹⁵. Consistent with the PCA results (**Fig. R5a**), top panel of **Fig. 1a** of our manuscript (also presented below as **Fig. R5b**) by SA show that the selected 30 cell lines are also clustered and separated into four quadrants of the two-dimension space. Similarly, we used SA for clustering the 1600 metabolic genes in the bottom panel of **Fig. 1a**, which yields an almost identical clustering. We have now added full description in the SI methods, section "**RNA-seq data processing, dimension reduction and clustering analysis**". In the text, we refer the reader to the Methods section. We reserve the elaboration of SA methodology for the spontaneous Raman spectra discussion, since that is where it is most novel and impactful.

Fig. R5. a) Dimension reduction analysis of expression profiles of 53 melanoma cell lines from **Fig. 1C** in the Cancer Cell paper¹² by PCA. b) Dimension reduction analysis of expression profiles of selected 30 melanoma cell lines (No RAS mutation) from **Fig. 1a** in our manuscript by SA.

We also appreciate the reviewer's careful counting for the points shown in the plots of our **Fig. 1a-b**. There are 30 points in the **Fig. 1a**. However, we did double-counted one cell line in the original **Fig. 1b**. We have updated **Fig. 1b** and modified the corresponding manuscript content.

Reviewer #2, Comment 2. *In Figure 1b, they present a heat map of cherry-picked genes from the 1600 genes. How was this choice made? Even if it is assumed that the authors consider only Fatty acid related, Amino-acid related, nucleotide-related and glycosylation-related genes, these would add up to several hundreds and not to 22 genes. The heat map provide a fundamental concern on the conclusions reached. Why not present a heat map of all the hundreds of genes involved in these processes, or even simply present a heat map of all fatty acid-related genes instead of using 7 genes. Figures 1 a and b are not credible and appear to be chosen to fit a hypothesis.*

Response: We thank the reviewer for this comment and sorry again for the confusion. The genes used for the heatmap in our original **Fig. 1b** were not cherry picked, but we did not describe sufficiently well how they were selected. In the previous shown **Fig. 1b**, the 8 phenotypic marker genes (top of heat map) were chosen based on literature-reported phenotypic markers known to delineate subtypes of melanoma cell lines^{16,17}. The 14 metabolic genes are the top ranked metabolic genes from an unbiased analysis, based upon contribution scores for each gene to the four phenotypes. The complete heatmap of top 100 metabolic genes from each of the four phenotypes (400 genes in total) is provided below as **Fig. R6** (also added as our new **Fig. S1**) and the corresponding list of the genes are provided in **Table R1** (added as our new **Table S1**). All transcriptome data are available in a public data base to permit additional analyses.

From the complete list of the genes (**Table R1**), the reviewer could see that the functions of each gene spans into different metabolic processes including fatty acid related, amino acid related, nucleotide related and sugar related metabolic processes. The previous selected 14 metabolic genes in the original **Fig. 1b** are color-coded in **Table R1**. As shown, they all rank at the very top of each mentioned metabolic process. To avoid any confusion, we have revised **Fig. 1b** to directly present top 4 ranked metabolic genes in each phenotype (total 16 genes). As the reviewer can see, this does not change our claim (manuscript, page 3) that "metabolic susceptibilities that exist within these cell lines may well vary with cellular de-differentiation". To make this point clearer, we also added in the text the following sentence (page 2): "Just like the well-reported phenotypic markers^{16,17}, metabolic genes also showed a clear phenotype-dependent expression trend, with associated functions that span different metabolic processes (The representative (top 4 ranked) metabolic genes are shown in the bottom of Fig. 1b, the complete heatmap and list of the top ranked metabolic genes are shown in Fig. S1 and Table S1)." With our above explanations together with the new **Fig. R6 (Fig. S1)** and **Table R1 (Table S1)** for the manuscript, we hope we can convince the reviewer that our heatmap is credible and does not provide a fundamental concern on the conclusions reached.

Fig. R6 Heatmap of gene expression levels for top 100 metabolic genes that are uniquely upregulated in each of the four phenotypes (400 genes in total) from an unbiased analysis of 1600 metabolic genes out of the whole transcriptomes of 30 melanoma cell lines.

Rank	S1 top 100	S2 top 100	S3 top 100	S4 top 100	Rank	S1 top 100	S2 top 100	S3 top 100	S4 top 100
1	TYR	RXYLT1	ST8SIA5	NNMT	51	PDE6B	GLUD2	HS6ST3	TYMP
2	DCT	ALDH1A1	GALNT5	MGST1	52	HSD17B8	LPL	NPR2	RDH10
3	GYG2	CYP27A1	DHRS3	GDA	53	QDPR	CERS1	BST1	PNMT
4	GMPR	NPR1	ALDH1A3	BCAT1	54	GCNT3	GUCY2C	BHMT2	AGPAT4
5	PNPLA4	ATP6V0A4	CYB5R2	MGAT5B	55	PFKFB2	PDE7B	CSGALNACT1	PDE9A
6	QPR1	GALNT5	NT5E	HS6ST3	56	CYP19A1	ASAH1	PDE7B	ALDOC
7	REBP	TYRP1	PLA2G7	MTAP	57	ST8SIA1	LDHD	GSTM3	PDE1C
8	ADCY2	UROC1	CHST1	GALNT14	58	GSTO2	FMO4	CYP27C1	CYP24A1
9	GALNT3	FOLH1	ACOX2	B3GALNT1	59	NAT8L	CKMT1B	PDE9A	NME7
10	GAPDHS	GAL3ST1	PDE1C	ANPEP	60	MGST1	EPHX2	AK5	MFNG
11	TYRP1	ST8SIA1	B3GNT7	PTGS2	61	RDH8	CSGALNACT1	AK6	DSE
12	PRDM7	B3GALT1	GALNT13	CHST15	62	GALNT12	TPK1	INMT	TXNRD1
13	ADCY1	TYR	B3GNT5	HS3ST3A1	63	HSD3B7	ALDH1A3	DDO	B3GNT3
14	PIP5K1B	DGK1	B3GALT2	HS3ST3B1	64	PIK3C2B	BAAT	ACSS3	PCK1
15	ALDOC	ENPP2	MGLL	CYP2S1	65	CKMT1A	PCYT1B	P4HA3	CA2
16	ATP6V0D2	ACP5	DPYD	PTGES	66	INPP4B	PDE2A	ABHD17C	NME3
17	BAAT	B3GNT7	AKR1C3	GUCY1A2	67	MAOA	SCD5	DGKA	B4GALT1
18	LPIN3	ALDH2	AKR1C1	NMNAT2	68	LFNG	ELOVL2	PLCE1	PGM2L1
19	PLB1	DCT	ST6GALNAC5	LFNG	69	ATP8	CRYL1	SYNJ2	CYP26A1
20	CA14	NT5E	TBXAS1	DPYD	70	CHAC1	B3GALT2	HSD17B3	SYNJ2
21	CHSY3	ST6GAL1	ELOVL4	ALPP	71	ALDH3B2	CKMT1A	NMRK1	PLCD3
22	PTGDS	ST8SIA5	ABAT	CYP1B1	72	PLCG2	CA12	PDE10A	ACOT4
23	LDHC	COLGALT2	MGST2	CDS1	73	NAT8	ENPP1	GFPT2	UGCG
24	TKTL1	CYP7B1	AKR1B10	GALNT6	74	NAT8B	ACOT12	COX7A1	AMPD3
25	OGDHL	ASPA	UGT8	AOX1	75	PDE11A	CYP19A1	ME1	GGT5
26	PKLR	POLR2F	PIK3CG	ASS1	76	GALM	PIK3CG	GAD1	KYNU
27	ACP5	EXTL1	HS3ST5	AKR1C3	77	ARG2	PRDM7	ACSM4	RRM2
28	CA8	REBP	PHGDH	PLA2G16	78	GSTA1	CNDP1	OPLAH	NMNAT3
29	UGT2B7	XYL1	LPL	B3GALT5	79	CKB	ABAT	DSE	NQO1
30	GLUL	ST3GAL6	EXTL1	ST6GAL2	80	PLD1	INPP5F	PLA2G4C	HS6ST2
31	PDE3B	MAN1C1	HEPH	GPX3	81	HSD17B6	GPD1	B3GAT1	EXT1
32	GUCY1A2	LARGE1	GPX7	MAN1A1	82	ACOT4	IL4I1	PIGZ	MBOAT7
33	CKMT1B	LARGE2	NMNAT2	XDH	83	PIP4K2A	MGAT5	HS3ST3A1	PDE6B
34	SELENO1	GAPDHS	CHST2	AK5	84	ST6GALNAC3	FADS2	AMT	PDE4D
35	ACSBG1	TBXAS1	PCYT1B	AK6	85	ISYNA1	DGKG	TUSC3	GLUL
36	GALC	PDE3B	ALDH2	INPP5J	86	1	HYAL1	FADS2	NSD1
37	PIK3CD	PDE3A	A4GALT	CPT1C	87	MAT1A	CHST11	ENPP4	NSD2
38	CYP1A1	CHST6	B4GALNT1	ENPP4	88	CYP24A1	BBOX1	UXS1	NSD3
39	ATP6V0A4	GSTM1	CHST7	CA8	89	RDH12	ACOX2	NOS2	ENTPD8
40	NPL	MAT1A	ANPEP	ACSL5	90	ACER2	GDPD1	ALDH3B1	TCIRG1
41	HNMT	MGAM	AOX1	PDE11A	91	PDE5A	ADCY2	XDH	CYP27C1
42	MGAT4A	MGAM2	FOLH1	HSD17B2	92	HYAL1	PIGZ	MFNG	HACD1
43	NMRK2	GALNT3	ETHE1	DGKE	93	HSD3B2	GLDC	IDS	PLOD2
44	B3GALT4	CA14	RXYLT1	PIK3CD	94	B3GALNT1	DDO	PLCH1	GCK
45	MTAP	ENPP3	VNN1	GBGT1	95	ASAH1	GNS	INPP5F	GSTM3
46	ST3GAL6	CHST7	PLOD2	GALC	96	MTMR8	PDXP	BCAT1	BDH1
47	FHIT	GSTA4	TPH1	PDE5A	97	LDHD	ACSL1	PGM3	ALDH1L2
48	PNLIPRP3	UGT2B7	CHST3	MOCOS	98	PDE4D	FHIT	GCNT1	ADK
49	HOGA1	HS3ST1	GALNT18	SPTLC3	99	HS6ST2	HOGA1	PTGES	IPMK
50	DGKE	MOGAT1	HS3ST1	ALPG	100	GSTO1	PNPLA4	A3GALT2	MARS

Table R1 The genes list in Fig. S1. S₁-S₄ columns each indicates the top 100 ranked metabolic genes uniquely upregulated in the melanocytic (S₁), transitory (S₂), neural crest (S₃) and undifferentiated (S₄) phenotypes in Fig. S1. The genes chosen for presentation in the original Fig. 1b are color-coded. Top 4-ranked genes in each column are now presented in the revised Fig. 1b.

Reviewer #2, Comment 3. *The idea of using Raman Spectra and the surprisal analysis are both exciting aspects of the manuscript. The authors have chosen 5 representative cell lines. What is the variance if more cell lines are chosen (this needs to be demonstrated)? How did they choose the 5 cell lines? How was the averaging done for each of the 10 cells/line?*

Response: The five cell lines studied were chosen based upon the single criteria that they uniformly span the four phenotypes of Fig 1a. We modified the sentence introducing the cell lines

studied in text (page 3): “We selected five representative patient-derived cell lines **based upon the single criteria that** they collectively spanned the range of de-differentiation status (indicated at the top of Fig. 1b with information listed in Table S2), from M381 (undifferentiated) to M262 (differentiated).” The reviewer does indicate that, in order to support claims of generality of our conclusions across BRAF mutant melanoma cell lines, we should measure more of the cell lines from the 30 of **Fig. 1a-b**. We have instead removed statements of generality from the text, and indicated that our results apply to the cell lines selected (although they are strongly suggestive of trends). We also added a few qualifying statements to the discussion section with respect to generality: “Another aspect that worth exploration is the generality of the cell line specific results reported here. For example, whether the susceptibility of SCD1, as revealed in the mesenchymal M381 cells, applies more generally across mesenchymal BRAF mutant melanoma tumors, is both intriguing and important, given the challenges in drugging such tumors.”

In our response to Reviewer #1, Comment 4 above, we have described that the variance in the data from spontaneous Raman imaging (**Fig. R3**), which is what the reviewer is referring to, arises from intracellular heterogeneity. We were able to prove this point of intracellular heterogeneity with SRS imaging data that is shown in the paper, in which the error bar is much smaller (**Fig. 2a-b** compared with **Fig. 1e-f**). Thus, it is not a matter of measuring more cell lines to reduce this variance, it is a matter of redesigning the instrument so that each spontaneous Raman spectrum averages over each single whole cell. We plan to custom build our spontaneous Raman spectrometer to perform reliable analysis with lower variance on more cell lines as our next-step goal. As indicated above, we have modified the method part “Spontaneous Raman spectroscopy for whole spectrum-wide cellular Raman spectra acquisition” in SI to reflect the choice of acquisition and averaging spontaneous Raman conditions: “To reduce spectral variance for spontaneous Raman spectra caused by intracellular heterogeneity, we recommend that at least three biological replicates with at least ten cells in each replicate are acquired for analysis.”

Reviewer #2, Comment 4. Assuming the variance is low and the deconvolution is reasonable, the authors arrive at CH₂ and CH₂ vibrations and see the differences between these across the cell line types. They conclude that CH₂/CH₃ ratio decreases with the progression of dedifferentiation. If the analysis is correct, this could even be a true observation. However, the conclusions they draw from this are not warranted without a significant number of control experiments. The authors conclude that elevated FASN activity in the more differentiated melanoma cell lines contributes to metabolic susceptibility. This conclusion was arrived based on the CH₂/CH₃ ratio. Alternate hypotheses would also fit such a result as described below. In the process of de-differentiation, there are significant histone modifications, involving demethylation, methylation, etc. Why would this not contribute to the CH₂/CH₃ ratio? An experiment like a genome wide ChIP-seq with H3K27/H3K27Ac and H3K4 could help in identifying the degree of methylation changes.

Response: We thank the reviewer for the comment on the CH₂/CH₃ ratios. For this comment, we want to point out that the trend observed for CH₂/CH₃ ratios is not simply from spontaneous Raman spectra analysis. The change of CH₂/CH₃ ratios with cell lines are confirmed by both sets of independent experiments from spontaneous Raman spectra on fixed cells (**Fig. 1c-1g**) and SRS imaging analysis on both live and fixed cells (**Fig. 2a-2b**). Although error bars are relatively high for the constraint from surprisal analysis of spontaneous Raman spectra, which, as indicated above in **Fig. R3**, is due to the intracellular heterogeneity and the small sampling number for spontaneous Raman spectra. In contrast, our high-resolution SRS imaging (**Fig. 2a-2b**) bypasses the intracellular heterogeneity issue and showed the same conclusion with straightforward data interpretation (i.e. no deconvolution). Each set of experiments shows statistically significant results and are confirmed by at least three experimental replicates. Therefore, we conclude the observation of CH₂/CH₃ trend is correct.

The implication from increased CH₂/CH₃ ratios for elevated FASN is also confirmed by both the gene expression data (**Fig. 2c-2d**), and the separate live-cell functional assay of lipid *de novo* synthesis with deuterated glucose¹⁸ (**Fig. 2e-2g**). We believe these evidences (i.e. Raman spectroscopy, Raman imaging, transcriptomics, activity assay) are corroborative and strong enough to warrant a true observation.

In addition, we appreciate the reviewer's suggestion of alternative explanation of histone modification such as methylation and demethylation that contribute to the change of CH₂/CH₃ ratios in different cell lines. Through analysis of current data and quantitative estimation below, we believe we could safely rule out this possibility. Our reasons are from two different perspectives and are listed below.

1) The spatial information, captured by SRS images (Fig. 2a), is crucial here. We can see that the difference of cellular CH₂/CH₃ ratios (**Fig. 2b**) across cell lines mostly originated from the difference of CH₂/CH₃ ratios in cytoplasm but not nucleus (**Fig. 2a**). To prove this point, we performed further quantification for CH₂/CH₃ ratios solely from the cytoplasm or from nucleus. The new results are summarized in **Fig. R7**. The data indicates that CH₂/CH₃ ratios of cytoplasm follow the same trend as that from the whole cells and are the primary contributors to the difference of whole-cell ratios across different cell lines (**Fig. R7a**). In contrast, the ratios from the nucleus are similar across cell lines (**Fig. R7b**). Since histone proteins predominantly exist in the nucleus and there is negligible change of CH₂/CH₃ ratios observed from the nucleus across cell-lines, histone methylation or demethylation should not contribute to the observed difference of CH₂/CH₃ ratios. This also shows the necessity of having high-resolution microscopy (i.e. our SRS imaging) for subcellular characterizations.

Fig. R7 Quantification of CH₂/CH₃ ratios of a) cytoplasm and b) nucleus from live single cells. n=30 cells, data plotted from min. to max.

2) We also theoretically calculated the maximum signal contribution to the change of cellular CH₂/CH₃ ratios from histone methylation or demethylation, and concluded the contribution is quite minimal.

There are about 10¹⁰ proteins (by extrapolating protein density reported in *E. Coli* to mammalian cells¹⁹) with an average length of 438 amino acids/protein in human cells¹. Among the amino acids, Thr (6.2 %), Met (1.8 %) and Ala (7.4%) have 1 methyl group, and Val (6.8 %), Ile (3.8 %) and

Leu (7.6 %) have 2 methyl groups. Considering their relative abundance (indicated in the brackets²⁰) in the human proteome, the total number of methyl groups from proteins in a human cell that yield the Raman signal at 2940 cm⁻¹ is estimated to be:

$$10^{10} \times 438 \times (6.2\% + 1.8\% + 7.4\% + 6.8\% \times 2 + 3.8\% \times 2 + 7.6\% \times 2) = 2.269 \times 10^{12}$$

As a comparison, there are about 3×10^7 chromosomes in a human cell²¹, and each chromosome has two sets of histone H2A, H2B, H3 and H4. Assuming all the possible histone methylation marks²² are completely methylated, then each chromosome will have $2 \times 129 = 258$ methyl groups (one set of histones H2A, H2B, H3 and H4 has a sum of up to 129 methyl groups estimated from previous reference²²). In total, a human cell has a maximum number of methyl groups for histone modification up to

$$3 \times 10^7 \times 258 = 7.74 \times 10^9.$$

The resulting theoretical ratio between the largest number of methyl groups derived from histone methylation and the total number of methyl groups in total proteins in a human cell is determined to be smaller than 0.5%:

$$\frac{7.74 \times 10^9}{2.269 \times 10^{12}} = 0.00341 = 0.34\%$$

This small percentage would have been easily buried in the noise of the data, especially in the spontaneous Raman spectra. We therefore conclude that histone methylation has negligible effect on the change of CH₂/CH₃ ratios across cell lines with above two reasonings.

Reviewer #2, Comment 5. The authors focus on fatty acid synthesis. How about alterations in other lipids? Did they examine the 2 major lipid processing pathways, one involving acyl CoA processes (considered by the authors) and the other involving acetyl CoA (not analyzed by the authors) involving cholesterol and sphingolipid pathways?

Response: The reviewer suggests that the possible involvement of other lipid processes in the lipid biosynthesis pathway might also explain the correlation of CH₂/CH₃ ratios with fatty acid synthesis. We want to first emphasize two points: 1) Fatty acid chains are the main constituents for all lipids. All subsequent lipid alterations are downstream of the synthesis of these fatty acids. Therefore, fatty acid synthesis is the dominant lipid synthesis pathways. 2) The CH₂ Raman signals (2845 cm⁻¹) dominantly come from the long fatty acyl chains of the lipids. Therefore, the spectral feature of both spontaneous Raman spectra analysis and CH₂/CH₃ ratios guide us to focus on long chain fatty acid synthesis pathways.

To better answer the reviewer's questions, we provide a more complete diagram of glucose-derived *de novo* lipid synthesis pathways (**Fig. R8**). Indeed, the acetyl-CoA can proceed with both the fatty acid synthesis pathway to synthesize fatty acids and the downstream products of lipids including sphingolipids; and the cholesterol synthesis pathway to generate cholesterol and cholesteryl ester (CE). There are several reasons for us to focus on fatty acid synthesis pathway. First, since our data revealed a highly linear correlation between CH₂/CH₃ ratios and the unbiased GESA pathway analysis scores for the fatty acid biosynthesis pathway (**Fig. 2d**), we hypothesized that fatty acid synthesis is a better candidate for revealing new susceptibilities due to the consistent trend of changes from both Raman and gene expressions across cell phenotypes. Second, the CH₂ signals at 2845 cm⁻¹ are from CH₂ in the long acyl chains of fatty acids. As a comparison, the CH₂ signal in cholesterol is blue shifted to around 2957-2997 cm⁻¹, indicated in **Fig. 3e**, due to the different chemical structure of sterol rings. It is hence less likely that the

cholesterol biosynthesis could have large influence on the CH₂/CH₃ ratios. Third, the relative abundance of cholesterol is relatively low. One major form of cholesterol in cells is CE. As shown in **Fig. R9**, the percentage of CE is about only 2%.

We therefore believe fatty acid synthesis pathway as the dominant pathway accounts for the change of CH₂/CH₃ ratios. It is further supported by the gene expression level of HMGCR (HMG-CoA reductase), the rate limiting enzyme in cholesterol synthesis pathway. Compared with the expression level of FASN, that of HMGCR is relatively low (**Fig. R10**). Moreover, there was no obvious correlation between CH₂/CH₃ ratios and the gene expressions for HMGCR (**Fig. R10**), in addition with a few other major enzymes (e.g. FDPS, SQS) involved in the cholesterol synthesis pathway. This further indicates that cholesterol synthesis pathway is not a major contribution to our observed trend of CH₂/CH₃ ratio across cell lines.

Fig. R8 *De novo* Lipid biosynthesis pathways for fatty acids and cholesterol through acetyl-coA.

Fig. R9 Complete bulk lipidomics of M381 cells. Total cholesteryl esters and sphingolipids account for 2.03% and 4.94% of full lipids, respectively. Phosphatidylcholine (PC), Phosphatidylethanolamine (PE), Free fatty Acids (FFA), Triacylglycerol (TAG), Diacylglycerol (DAG), Lysophosphatidylcholine (LPC), Lysophosphatidylethanolamine (LPE), Cholesterol Ester (CE), Sphingomyelin (SM), Ceramides (CER), Dihydroceramides (DCER), Hexosylceramides (HCER), Lactosylceramide (LCER). n=3, data shown as mean \pm SEM.

Fig. R10 Gene expression levels for FASN and HMGCR across cell lines.

Reviewer #2, Comment 6. Further the authors in the cell prep use FCS which contains fatty acids including arachidonic acid which is processed in the cells to produce fatty acyl derivatives (in fact the increase in Phospholipase gene levels may point to this processing). What controls were used to eliminate this confounding effect?

Response: We thank the reviewer for this question. We believe the reviewer is interested in the activities of fatty acid uptake. Although we did not control the level of FBS in the cell medium, we actually have explored this question as shown in **Fig. S9a-d**. We cultured all five cell lines in medium with supplied relatively low concentration (50 μ M) of deuterated fatty acids (i.e. palmitic acids and oleic acids, the primary saturated and unsaturated fatty acids in FBS²³). We did find that the overall trend of fatty acid uptake is similar to that of fatty acid synthesis across cell lines. This means that both fatty acids synthesis and fatty acids uptake could contribute to the higher lipid pool in more differentiated cell lines. However, it is not a confounding effect, and doesn't interfere with our associated claims, which are 1) More differentiated cell lines have higher CH₂/CH₃ ratio. 2) Fatty acid synthesis pathway is also more active in the more differentiated cell lines. 3) Fatty acid synthesis pathway is a druggable susceptibility to more differentiated cell lines.

The reviewer also points out the increase of gene expression of phospholipase gene, PLB1 in **Fig. 1b**. In addition to analyzing their relative gene expression trend as in **Fig 1b**, we now compare the absolute expression levels of PLB1 and FASN in five cell lines. As shown below in **Fig. R11**, the absolute expression levels of PLB1 are 300-400 folds below the expression levels of FASN in the same cells, which indicates that this pathway is much less active compared to the FASN pathway.

Fig. R11 Gene expression levels for FASN and PLB1 across cell lines.

Reviewer #2 Comment 7. The authors describe that M381 phenotype accumulates more unsaturated lipids and cholesteryl esters in lipid droplets. They describe their prior work on lipid peroxidation in mesenchymal-specific GPX4-inhibitor induced ferroptosis. What is the cross talk between oxidative processes and lipid saturation. The GC-MS results are shown without details. What standards were used in quantification of the fatty acids. Figure 3 states that percentages of 16:0, 16:1, 18:0, 18:1, 18:2, and 20:4 are normalized to all extracted fatty acids (n=4-5). More details would help in understanding how the relative quantification was achieved.

Response: We thank the reviewer for the interest in correlating these two processes. In general, unsaturated lipids are more prone towards oxidization and generating peroxides, which may induce oxidative stress, including ferroptosis. However, the main reason that we mention the prior work on lipid peroxidation in mesenchymal-specific GPX4-inhibitor induced ferroptosis before discussing our results of lipid unsaturation is to emphasize the importance of related lipid processes in the mesenchymal phenotype. A distinct difference between the current and prior work is that our analysis of lipid unsaturation is at the subcellular level, specifically from the lipid droplets instead of whole cells. The subcellular enrichment of unsaturated lipids in lipid droplets is not obvious in bulk or even whole cell analyses (**Fig. 4g and 4h**).

For GC-MS analysis on lipids, we followed the standard protocol²⁴. As indicated in SI, “Peaks were identified by comparing the mass spectra and retention times to the authentic standards and library data. Quantification was achieved by a flame ionization detector. Absolute concentrations are determined with reference to internal standards.” However, the absolute concentrations are influenced by sample loss in the preparation steps of lipid extraction and purification. We therefore present the data as relative percentage for each lipid component. We have added the following sentences in the SI to better explain the details: “To avoid complications from sample loss at sample preparation stage, we used the relative abundance of each species of fatty acids for data interpretation. Relative abundances were calculated by dividing the peak area for each of the six most abundant fatty acids (16:0, 16:1, 18:0, 18:1, 18:2, and 20:4) to the sum of peak areas of all six species. The signals of other species are too low and mostly buried in noise.”

Reviewer #2, Comment 8. In discussing lipid desaturases, the authors point to their involvement in elevated lipid-droplet unsaturation of the M381 mesenchyme phenotype. Did the authors control for PUFA in the media? How were quantifications achieved? In Figure 4d were relative viability studies carried out for longer time to assess if lines other than M381 also showed

apoptotic effects? This is important to the conclusion that CAY inhibition of SCD1 for MUFA synthesis leads to more significant viability loss for M381 cell lines.

Response: We believe that the reason the reviewer is asking whether we controlled for PUFA in the media is to see if M381 uptakes larger amount of PUFA, which is shuttled to lipid droplets and therefore shows higher unsaturation. All cells were cultured in regular growth medium and we did not control for PUFA in the media. We reasoned that the uptake of PUFA is not the major source for elevated unsaturation in lipid droplets for M381 cells through quantification of SRS signals and the lipidomics data described below.

First, we used the intensity ratio of $3022\text{ cm}^{-1}/2908\text{ cm}^{-1}$ in hSRS spectra specifically obtained from lipids droplets to quantify the relative number of C=C bonds from lipids in the droplet (**Fig. 3f-g**). The 3022 cm^{-1} peak represents the C-H bonds directly attached to the C=C bonds while the 2908 cm^{-1} intensity represents the overall C-H. As shown in **Fig. 4b-c**, we quantified the intensity of $3022\text{ cm}^{-1}/2908\text{ cm}^{-1}$ ratios after incubating cells with CAY (SCD1 inhibitor) or SC ($\Delta 6$ desaturase inhibitor). The decrease of the $3022\text{ cm}^{-1}/2908\text{ cm}^{-1}$ ratios indicates the existence of both MUFA and PUFA in the droplets. While each PUFA has at least two C=C bonds, the CAY treatment results in much larger decrease for the unsaturation ratio than SC (**Fig. 4b**). This implies that MUFA is the dominant species in the droplets. Second, we performed new analysis from our lipidomic data of M381 cells (**Fig. S11**) for the total MUFA and PUFA percentages in TAG and CE, the resident species in the lipid droplets. TAG contains 55% of MUFA and only 10% of PUFA; and CE has 62% of MUFA and only 16% of PUFA. This again confirms that MUFA is the major lipid species in the droplets.

In FBS, lipoproteins, especially low-density lipoproteins (LDL) are the primary carrier of PUFA into the cells²⁵. We first performed new measurements of LDL uptake by fluorescence imaging of Dil-LDL to directly quantify the ability of PUFA uptake across cell lines (**Fig. R12**). As shown in **Fig. R12a**, M381 has lower uptake of LDL (red fluorescence) than other cell lines. This is further supported by quantification of fluorescence signals per cell (**Fig. R12b**). In addition, to test the potential influence for M381 cells from PUFA uptake, we performed a set of new experiments of viability assays for all cell lines cultured in medium containing lipoprotein deficient serum (LPDS, **Fig. R13**). We expect that M381 cells would have higher viability when cultured in medium without lipoproteins (i.e. LPDS) compared to the other cell lines if M381 cells since this cell line takes up less PUFA. Indeed, as shown in **Fig. R13** below, M381 cells maintain relative high viability in the LPDS medium, confirming that M381 cells don't rely more on PUFA uptake in the medium compared to other cell lines. These two evidences (**Fig. R12 & R13**) prove that the PUFA uptake from medium is not the primary reason for the elevated unsaturation of lipid droplets in M381 cells and doesn't affect our claims.

Fig. R12 LDL uptake assay cross cell lines. a) Representative fluorescence images of melanoma cells' uptake of Dil-LDL. Grey: CH₃ SRS; Red: Dil fluorescence. Scale bar: 20 μm . b)

Quantification of average Dil-LDL fluorescence signals per cell across cell lines (5 frames, n>50 cells, data shown as mean \pm SEM).

Fig. R13 Viability tests of melanoma cell lines cultured with medium containing lipoprotein deficient serum (LPDS) for 3 days. n=4, data shown as mean \pm SEM.

In light of the reviewer’s suggestion for longer-time assessment of CAY effect, we performed an extended time-dependent (from 2 to 5 days) viability assay with drug CAY across cell lines (**Fig. R14**). The results clearly show that CAY inhibition of SCD1 leads to much more significant viability loss for M381 cells even in a longer treatment period. This data is now included as new **Fig. S12**.

Fig. R14 Viability assays of melanoma cell lines under different days of 1 μM CAY treatment. a) Dependence of viability for cell lines on the treatment length with 1 μM CAY. b-e) Bar-chart plot for comparing viability in (a) across five cell lines after 2-day (b), 3-day (c), 4-day (d), and 5-day (e) of CAY treatment. n=4, data shown as mean \pm SEM.

Reviewer #2 Comment 9. *While the broad conclusions are interesting and perhaps can be useful for drug targeting, their validity should be demonstrated through more rigorous control experiments.*

Response: We hope we have fully alleviated the reviewer’s concern by clarifications of our results and demonstrations of new results.

Response to Reviewer #3:

We thank the reviewer for all the comments that would help solidify our paper and broaden its impact to the metabolic field. We particularly appreciate the reviewer's recognition that our work demonstrates "a significant technique" to "monitor metabolic changes in single live cells"; and "Overall this is an impressive study. The detection of drug susceptibility of the SCD1 pathway in the Raman data, not detected in transcriptomic or through other metabolomic techniques, validates a new approach to studying metabolic changes relevant to cancer." The reviewer also raised some minor comments on the technical part of our methods. Below, we give a point-by-point response to the reviewer's comments.

Reviewer #3 Comment 1. *The manuscript suggests that Raman can be broadly applied to monitor metabolic changes in cells. The use of Raman scattering to detect lipid droplets is well established as cited in the manuscript (e.g. Ref. 22). In particular, the high concentration of lipids confined in lipid droplets are readily detectable in Raman experiments. Can the authors comment on quantitative metrics associated with metabolic changes? For example, what concentration is necessary for detection? It is not clear this methodology can be widely applied beyond the CH vibrations detected with lipids or with isotope tags (i.e. the deuterium incorporation through D-labeled glucose metabolism) that exist at otherwise vibrationally quiet frequencies.*

Response: We understand the reviewer's concern of more general applicability beyond the CH vibrational region due to the sensitivity limit. Below we give more specific numbers to clarify for the quantitative metrics.

For label-free imaging, the SRS detection limit is 50 μM and 5 mM for retinol and methanol respectively, in a similar set-up as ours⁶. This translates into bond-specific detection limit of 50 μM for conjugated double bond in retinol and 15 mM of C-H bond. Considering that palmitic acid (PA) has 31 C-H bonds, and oleic acid (OA) has 34 C-H bonds, their detection limits are estimated to be 480 μM and 440 μM , respectively. Similarly, the SRS imaging sensitivity for deuterated PA (PA-d₃₁) and OA (OA-d₃₄) were recently determined to be 480 μM and 450 μM , respectively²⁶. This number may indicate a low sensitivity for molecular-specific detection. However, it is more than enough for imaging many chemical species with generally above mM concentrations (e.g. proteins, lipids) in the cells. For example, SRS imaging of C-D bond (with similar cross section to C-H bond) can be used to detect palmitoyl groups from a single lipid bilayer of Dipalmitoylphosphatidylcholine (DPPC)⁸. Although the sensitivity of SRS has not reached the level of fluorescence imaging, the label-free imaging or minimally perturbative isotopic labeling aspect of SRS avoids the perturbation from large fluorophore labeling mostly required for fluorescence microscopy. This makes SRS a more ideal tool to probe the activity for small metabolites in cells.

Going beyond the C-H spectral window, a large number of metabolites have been investigated by SRS or spontaneous Raman imaging targeting the vibrational features in the fingerprint region or in the cell-silent window with C-D or alkyne tags. Metabolites demonstrated for cellular imaging in the fingerprint region include acetyl-choline (720 cm^{-1})²⁷, cytochrome C (750 cm^{-1})²⁸, Guanine and adenine (1340, 1490, 1580 cm^{-1})²⁹, tyrosine and tryptophan (1615 cm^{-1})²⁹, DNA (788 cm^{-1})³⁰, RNA (813 cm^{-1})³⁰, glycogen (860, 938 cm^{-1})³¹. In the silent region (1800-2700 cm^{-1}), the labeling tags could go beyond C-D tags to triple bonds, including alkyne and nitrile tags. One main merit of these Raman tags compared to isotopes is the higher Raman cross section. This allows imaging for a broad spectrum of metabolites with a higher detection sensitivity. For example, EdU (5-ethynyl-2'-deoxyuridine) a widely used thymidine analogue, can be used to probe newly

synthesized DNA in live cells and tissues with a detection limit of 200 μM ³². Another example is that the phenyl-diyne tagged cholesterol can be used to probe cholesterol transport and storage in live cells and *C. elegans* with a detection limit down to 31 μM ³³. More numbers regarding sensitivity and applications with Raman tags have been summarized in a recent review³⁴.

Reviewer #3 Comment 2. *It appears the cell results reported are determined from fixed cells. This has real significance for live cell studies. What is the potential for live cell imaging? How does the variance from individual cells compare to the trends observed in the averaged data presented? How many cells are necessary in such experiments?*

Response: We agree with the reviewer that the live-cell compatibility of SRS has real significance. Indeed, the biocompatibility is a notable advantage of SRS imaging, which has been widely demonstrated in a number of recent publications for imaging live cells³², live tissues³⁵, live animals ranging from *C.elegans*³⁶, rodents³⁷ to humans³⁸. We, in fact, also demonstrated live-cell SRS imaging and quantification in our **Fig. 2a**. The reason that we performed most of our hSRS quantification for lipid droplets in fixed cells was because our laser has relative slow tuning between wavelengths. This would add complication for data analysis and would possibly cause motion artifact for acquiring a stack of hyperspectral images in live cells. However, this is an instrumentation limit of our current set-up, but not a fundamental limit. With recent development of spectral-focusing SRS imaging³⁹, hyperspectral imaging of live cells (similar to the SI video) can be acquired in seconds.

For the questions regarding the variance from individual cells. We have provided more detailed response regarding the variance from spontaneous Raman to Reviewer#1 Comment 4 (Fig. R3). Briefly, the relatively large variance in spontaneous Raman was due to the intracellular heterogeneity and the relative low sampling for spontaneous Raman spectra. We thus recommend that more replicates should be repeated. We have added to the SI the following sentence: “To reduce spectral variance for spontaneous Raman spectra caused by intracellular heterogeneity, we recommend that at least three biological replicates with at least ten cells in each replicate are acquired for analysis.”

With much higher resolution, faster acquisition (hence much larger sampling) that bypass the intracellular heterogeneity, the variance of SRS imaging across individual cells is much smaller (**Fig. 2a-b** compared with **Fig. 1e-f**). Besides, to ensure day-to-day performance of our SRS system, we use signals from pure deuterated water for system calibration. For SRS imaging, we indicated in the SI that, “We recommend that at least three biological replicates with at least five cells in each replicate are acquired for analysis.”

Response to Reviewer 4:

We thank this reviewer for all the constructive and insightful comments to improve our paper and suggest a future direction. We particularly appreciate the reviewer's recognition that our work "provides and interesting proof of concept for the use of Raman spectro-microscopy in identifying phenotype-dependent metabolites in living cells at the singly cell level." Meanwhile, we understand that the reviewer hope to see more potential and benefits of our methods. We intend to argue that both our method and our identified targets as well as illustrated metabolic control underlying them are novel to the melanoma field. We added the future directions in the discussion section that we intend to explore in order to show more potentials and benefits of our method to melanoma studies and to cancer field in general. Below, we would like to give a point-by-point response to the reviewer's comments.

Reviewer #4 Comment 1. *The authors provide a nice proof of principle of the use of Raman technologies for probing metabolites at the single cell level. Unfortunately, restricting to rather uniform cell line cultures, the authors fail to show the full potential and benefit of these approaches for single cell analysis. I would argue that the targets that have been identified could also have been identified using standard lipidomics approaches on bulk. Inclusion of a more heterogenous model system would have substantially augmented the merit of this paper.*

Response: We thank the reviewer for recognizing the high potential of our method. We agree that the full potential of our method could be further demonstrated in the more heterogeneous tissue sample. This is exactly our next step, as indicated in our discussion. We would argue that our proof-of-concept study here clearly illustrated the benefits of single-cell, or even subcellular Raman analysis for guiding pharmaco-metabolomics study even in these relatively uniform cell samples. The identified targets, in particular, the lipid unsaturation, would not be readily revealed by standard methods, such as bulk lipidomics. This is discussed extensively in the manuscript, with supporting data.

For the target of lipid mono-unsaturation, as we stated in the paper, the elevated-level of lipid unsaturation in lipid droplets (**Fig. 3f-g**) is averaged out in the bulk measurements for whole-cell fatty-acid unsaturation (**Fig. 3g**). The reviewer may argue that if we did whole cell lipidomics measurements across cell lines, the fatty acid compositions may reflect that M381 cells have high unsaturation in TAG and CE. However, TAG and CE only accounts for 3.66 % and 2.03 % of total lipids as in M381 cells. It is not reasonable to focus on these two types of lipids purely based on lipidomics without any guiding evidence. In contrast, our further analysis of TAG and CE in lipidomics was guided by chemical composition of spatially defined lipid droplets revealed from hSRS imaging. In addition, our Raman microscopy offers much higher throughput and is more cost-effective per sample compared to lipidomics. Objectively speaking, Raman spectro-microscopy would not be able to replace traditional metabolomic analysis like lipidomics, but to offer complementary information to the existing methods to guide previously non-explored aspects of research.

Reviewer #4 Comment 2. *Along the same lines, the concepts and targets that have been identified may not be as novel as the authors claim. De novo lipogenesis have been extensively documented as potential target in melanoma (Talebi et al., 2018; Wu et al., 2018;...). Also, SCD1 is a known target in studies on melanoma (e.g. Pisanu et al, 2018). The involvement of detergent-resistant microdomains and lipid droplets as a possible reservoir for unsaturated fatty acids have also been reported.*

Response: We thank the reviewer for this comment, which would allow us to better clarify our novelty claim. We want to point out that the susceptibilities and the mechanism underlying these susceptibilities in the papers that the reviewer indicated are distinctively different from what we identified. First, the question we are tackling is different from above-mentioned papers. Second, along the same line as our response above, we have subcellular metabolic information to guide our further study. Such subcellular information would have been easily averaged out in the bulk analysis (indicated in our **Fig. 3g**), which is the focus in those papers.

To be more specific, in the Talebi *et al.* 2018⁴⁰, the researchers investigated the melanoma cells' response to BRAFi, and studied heterogeneous metabolic responses during BRAFi treatment across cell lines. They then identified sustained SREBP-1-dependent lipogenesis as a key mediator of resistance to BRAF-targeted therapy. Both the question and the identified target are vastly different from our report here. First, they were studying cellular response to BRAFi treatment and we focused on metabolic studies for different melanoma phenotypes. Second, the target they identified is lipogenic SREBP1 but not FASN, identified by us.

Similarly, Wu *et al.*⁴¹ relied on gene expression analysis and found that BRAFi resistant melanoma cells have increased expression level of SREBP1-dependent *de novo* fatty acid synthesis genes during BRAFi treatment. They also found that inhibiting the *de novo* fatty acid synthesis pathway actually exerts potent cytotoxic effects on both BRAFi-sensitive and BRAFi-resistant melanoma cells. The methodology, the findings, the mechanism underlying the cytotoxic effects and the conclusion are all quite different from ours.

As for SCD1, Pisanu *et al.* 2018⁴² reported an increase in SCD1 expression during melanoma progression. As a comparison, our observation of increased unsaturation in lipid droplets does not associate with an increase of SCD1 expression for the M381 cells. In fact, M381 cells show the lowest SCD1 expression among all examined cell lines (**Fig. S8c**). The mechanism of metabolic regulation underlying the SCD1 susceptibility for M381 cells is utterly different from that reported by Pisanu *et al.* By the same criteria used in this previous report, M381 cells would have been categorized to be insensitive to SCD1 treatment. However, this is not the case.

For detergent-resistant microdomains, to the best of our knowledge, our paper is the first report for the formation of such detergent-resistant membrane structures under SCD1 inhibition in aggressive cancer cells. The previous paper, as we indicated in our manuscript, was for studying lipotoxicity and the authors only observed the formation of detergent-resistant microdomains in HeLa cells that are supplied with extra high-concentration (~ 0.5 M) saturated fatty acids in the culture medium. The direct transfer of the knowledge here is not feasible.

In summary, we would argue that our demonstrated methodology, the drugging susceptibility and the mechanistic picture integrating all pieces together are all novel in understanding the metabolism of melanoma cell lines of different phenotypes.

Reviewer #4 Comment 3. Cerulenin cannot be used as a bona fide inhibitor of fatty acid synthesis. Cerulenin is notoriously promiscuous and affects several other targets. Nowadays, more selective inhibitors such as TVB-3166 or IPI-9119 are available.

Response: We thank the reviewer for the suggestion of using more selective FASN inhibitor. Although cerulenin is not as selective as TVB-3166 or IPI-9119, it is still widely used in research as an effective inhibitor for fatty acid synthesis⁴³⁻⁴⁵. In light of the reviewer's suggestion, we performed additional viability assay with TVB-3166 (**Fig. R15**) since IPI-9119 is more difficult to obtain. We used the same concentration (0.2 μ M) as in previous report⁴⁶. The result shows the

same trend as cerulenin: more differentiated cell line has higher viability loss, which confirms our first druggable target of fatty acid *de novo* synthesis. We have added the data as our new **Fig. S6d**.

Fig. R15 Viability tests of melanoma cell lines under 0.2 μM TVB-3166 treatment for 3 days. n=4, data shown as mean ± SEM.

Reviewer #4 Comment 4. In Fig 1c, the intensity scale in the Y axis is confusing as the spectra are offset apart for presentation.

Response: We thank the reviewer for pointing this out, which helps make our data presentation clearer. The intensity scale is arbitrary unit (A.U.). The spectra are manually offset apart to be more distinguishable, without no scaling of intensities. The absolute heights of spectra can hence be directly compared, although they are vertically offset apart. This is a widely adopted spectra presentation^{32,47–50}. To avoid any confusion, we added a brief statement in the figure caption “Each spectrum is offset apart in y-axis with no changes of absolute intensities.”.

Reference:

1. Kozłowski, L. P. Proteome-pl: Proteome isoelectric point database. *Nucleic Acids Res.* **45**, D1112–D1116 (2017).
2. Movasaghi, Z., Rehman, S. & Rehman, I. U. Raman spectroscopy of biological tissues. *Appl. Spectrosc. Rev.* **42**, 493–541 (2007).
3. Clemens, G., Hands, J. R., Dorling, K. M. & Baker, M. J. Vibrational spectroscopic methods for cytology and cellular research. *Analyst* **139**, 4411–4444 (2014).
4. Wei, L., Yu, Y., Shen, Y., Wang, M. C. & Min, W. Vibrational imaging of newly synthesized proteins in live cells by stimulated Raman scattering microscopy. *Proc. Natl. Acad. Sci. U. S. A.* **110**, 11226–11231 (2013).
5. Wei, L. *et al.* Live-cell bioorthogonal chemical imaging: stimulated Raman scattering microscopy of vibrational probes. *Acc. Chem. Res.* **49**, 1494–1502 (2016).
6. Freudiger, C. W. *et al.* Label-free biomedical imaging with high sensitivity by stimulated Raman scattering microscopy. *Science* **322**, 1857–1860 (2008).
7. Listenberger, L. L. *et al.* Triglyceride accumulation protects against fatty acid-induced lipotoxicity. *Proc. Natl. Acad. Sci.* **100**, 3077–3082 (2003).
8. Shen, Y. *et al.* Metabolic activity induces membrane phase separation in endoplasmic reticulum. *Proc. Natl. Acad. Sci.* **114**, 13394–13399 (2017).
9. Kobayashi-Kirschvink, K. J. *et al.* Linear Regression Links Transcriptomic Data and Cellular Raman Spectra. *Cell Syst.* **7**, 104–117.e4 (2018).
10. Yen, C.-L. E., Stone, S. J., Koliwad, S., Harris, C. & Farese, R. V. DGAT enzymes and triacylglycerol biosynthesis. *J. Lipid Res.* **49**, 2283–2301 (2008).
11. Wang, L. *et al.* Structure and Catalytic Mechanism of Human Diacylglycerol O-Acyltransferase 1. *Nature* (2020). doi:10.1016/j.bpj.2019.11.2891

12. Tsoi, J. *et al.* Multi-stage differentiation defines melanoma subtypes with differential vulnerability to drug-induced iron-dependent oxidative stress. *Cancer Cell* **33**, 890–904 (2018).
13. Zadran, S., Arumugam, R., Herschman, H., Phelps, M. E. & Levine, R. D. Surprisal analysis characterizes the free energy time course of cancer cells undergoing epithelial-to-mesenchymal transition. *Proc. Natl. Acad. Sci.* **111**, 13235–13240 (2014).
14. Su, Y. *et al.* Phenotypic heterogeneity and evolution of melanoma cells associated with targeted therapy resistance. *PLoS Comput. Biol.* **15**, 1–22 (2019).
15. Bogaert, K. A. *et al.* Surprisal analysis of genome-wide transcript profiling identifies differentially expressed genes and pathways associated with four growth conditions in the microalga *Chlamydomonas*. *PLoS One* **13**, 1–22 (2018).
16. Su, Y. *et al.* Multi-omic single-cell snapshots reveal multiple independent trajectories to drug tolerance in a melanoma cell line. *Nat. Commun.* **11**, 1–12 (2020).
17. Müller, J. *et al.* Low MITF/AXL ratio predicts early resistance to multiple targeted drugs in melanoma. *Nat. Commun.* **5**, (2014).
18. Zhang, L. *et al.* Spectral tracing of deuterium for imaging glucose metabolism. *Nat. Biomed. Eng.* **3**, 402–413 (2019).
19. Milo, R. What is the total number of protein molecules per cell volume? A call to rethink some published values. *BioEssays* **35**, 1050–1055 (2013).
20. King, J. L. & Jukes, T. H. Non-Darwinian Evolution. *Science* **164**, 788–798 (1967).
21. Telser, A. Molecular Biology of the Cell, 4th Edition. *Shock* (2002).
doi:10.1097/00024382-200209000-00015
22. Huang, H., Lin, S., Garcia, B. A. & Zhao, Y. Quantitative proteomic analysis of histone modifications. *Chem. Rev.* **115**, 2376–2418 (2015).
23. Gregory, M. K. *et al.* Development of a fish cell culture model to investigate the impact of fish oil replacement on lipid peroxidation. *Lipids* **46**, 753–764 (2011).
24. Rodríguez-Ruiz, J., Belarbi, E. H., Sánchez, J. L. G. & Alonso, D. L. Rapid simultaneous lipid extraction and transesterification for fatty acid analyses. *Biotechnol. Tech.* **12**, 689–691 (1998).
25. Habenicht, A. J. R. *et al.* The LDL receptor pathway delivers arachidonic acid for eicosanoid formation in cells stimulated by platelet-derived growth factor. *Nature* **345**, 634–636 (1990).
26. Li, X. *et al.* Quantitative Imaging of Lipid Synthesis and Lipolysis Dynamics in *Caenorhabditis elegans* by Stimulated Raman Scattering Microscopy. *Anal. Chem.* **91**, 2279–2287 (2019).
27. Fu, D., Yang, W. & Xie, X. S. Label-free imaging of neurotransmitter acetylcholine at neuromuscular junctions with stimulated Raman scattering. *J. Am. Chem. Soc.* **139**, 583–586 (2017).
28. Okada, M. *et al.* Label-free Raman observation of cytochrome c dynamics during apoptosis. *Proc. Natl. Acad. Sci. U. S. A.* **109**, 28–32 (2012).
29. Kumamoto, Y., Taguchi, A., Smith, N. I. & Kawata, S. Deep ultraviolet resonant Raman imaging of a cell. *J. Biomed. Opt.* **17**, 0760011 (2012).
30. Uzunbajakava, N. *et al.* Nonresonant confocal Raman imaging of DNA and protein distribution in apoptotic cells. *Biophys. J.* **84**, 3968–3981 (2003).
31. Pascut, F. C. *et al.* Noninvasive detection and imaging of molecular markers in live cardiomyocytes derived from human embryonic stem cells. *Biophys. J.* **100**, 251–259 (2011).
32. Wei, L. *et al.* Live-cell imaging of alkyne-tagged small biomolecules by stimulated Raman scattering. *Nat. Methods* (2014). doi:10.1038/nmeth.2878
33. Lee, H. J. *et al.* Assessing Cholesterol Storage in Live Cells and *C. elegans* by Stimulated Raman Scattering Imaging of Phenyl-Diyne Cholesterol. *Sci. Rep.* **5**, 1–10

- (2015).
34. Shen, Y., Hu, F. & Min, W. Raman Imaging of Small Biomolecules. *Annu. Rev. Biophys.* **48**, 347–369 (2019).
 35. Wei, L. *et al.* Imaging complex protein metabolism in live organisms by stimulated raman scattering microscopy with isotope labeling. *ACS Chem. Biol.* **10**, 901–908 (2015).
 36. Fu, D. *et al.* In vivo metabolic fingerprinting of neutral lipids with hyperspectral stimulated raman scattering microscopy. *J. Am. Chem. Soc.* **136**, 8820–8828 (2014).
 37. Lu, F. K. *et al.* Label-free DNA imaging in vivo with stimulated Raman scattering microscopy. *Proc. Natl. Acad. Sci. U. S. A.* **112**, 11624–11629 (2015).
 38. Liao, C. S. *et al.* In Vivo and in Situ Spectroscopic Imaging by a Handheld Stimulated Raman Scattering Microscope. *ACS Photonics* **5**, 947–954 (2018).
 39. Alshaykh, M. S. *et al.* High-speed stimulated hyperspectral Raman imaging using rapid acousto-optic delay lines. *Opt. Lett.* **42**, 1548 (2017).
 40. Talebi, A. *et al.* Sustained SREBP-1-dependent lipogenesis as a key mediator of resistance to BRAF-targeted therapy. *Nat. Commun.* **9**, 1–11 (2018).
 41. Wu, S. & Näär, A. M. SREBP1-dependent de novo fatty acid synthesis gene expression is elevated in malignant melanoma and represents a cellular survival trait. *Sci. Rep.* **9**, 1–17 (2019).
 42. Pisanu, M. E. *et al.* Inhibition of Stearoyl-CoA desaturase 1 reverts BRAF and MEK inhibition-induced selection of cancer stem cells in BRAF-mutated melanoma. *J. Exp. Clin. Cancer Res.* **37**, 1–17 (2018).
 43. Gao, S. *et al.* Important roles of brain-specific carnitine palmitoyltransferase and ceramide metabolism in leptin hypothalamic control of feeding. *Proc. Natl. Acad. Sci. U. S. A.* **108**, 9691–9696 (2011).
 44. Cheng, G. *et al.* Cerulenin Blockade of Fatty Acid Synthase Reverses Hepatic Steatosis in ob/ob Mice. *PLoS One* **8**, (2013).
 45. Vadia, S. *et al.* Fatty Acid Availability Sets Cell Envelope Capacity and Dictates Microbial Cell Size. *Curr. Biol.* **27**, 1757-1767.e5 (2017).
 46. Ohol, Y. M., Wang, Z., Kemble, G. & Duke, G. Direct inhibition of cellular fatty acid synthase impairs replication of respiratory syncytial virus and other respiratory viruses. *PLoS One* **10**, 1–20 (2015).
 47. Yue, S. *et al.* Cholesteryl ester accumulation induced by PTEN loss and PI3KAKT activation underlies human prostate cancer aggressiveness.pdf. *Cell Metab.* **19**, 393–406 (2014).
 48. Hu, F. *et al.* Supermultiplexed optical imaging and barcoding with engineered polyynes. *Nat. Methods* **15**, 194–200 (2018).
 49. Curro, N. J. *et al.* Unconventional superconductivity in PuCoGa5. *Nature* **434**, 622–625 (2005).
 50. Dundas, C. M. *et al.* Exposed subsurface ice sheets in the Martian mid-latitudes. *Science* **359**, 199–201 (2018).

REVIEWERS' COMMENTS:

Reviewer #1 (Remarks to the Author):

Raman-guided Subcellular Pharmacology-Metabolomics for Metastatic Melanoma

Du, et al.

NCOMMS-20-02898-T (Revision)

My main concerns with the first version of this manuscript focused on the interpretation of Raman spectral images, especially in light of line shape analysis and advanced multivariate statistical analysis of the image data. My major comments addressed: (1) assignment of the complex SRS line shape in the C-H stretching region, specifically assigning an unresolved feature at 2845 cm⁻¹ to lipids and a similar feature at 2940 cm⁻¹ to protein; (2) the underutilization of surprisal analysis constraint values; (3) delineation between the discovery-science portions of the investigation and the more hypothesis-driven parts. In all cases the authors have undertaken additional experiments, or calculations, or both to assess the importance of the issues I raised. Given their careful and thoughtful approach to the comments made to the first version, I am now completely satisfied that the paper is both significant - in its major conclusions about the application of Raman microspectral imaging - and rigorous in its interpretation of a rich set of experimental data. I support publication in its current form.

Reviewer #2 (Remarks to the Author):

I thank the authors for addressing almost every question that arose. My only minor concern lies with the comment 3 posed initially on validity in only 5 cell lines. While the generality of the conclusion cannot be assumed, the authors state this explicitly and this reviewer agrees with the additional sentence.

Reviewer #3 (Remarks to the Author):

This is very interesting manuscript. The ability to monitor and spatially locate chemical variation to address metabolic changes and drug susceptibilities is quite exciting. The authors have nicely used previously established biochemical markers in the literature (CH₂/CH₃ ratios to correlate lipid and protein rich regions) and isotopic tags to follow the changes in metabolism. The article opens new avenues opens new ways to monitor metabolism in cells. I support publication of the revised manuscript.